# Synthetic circuits reveal how mechanisms of gene regulatory networks constrain evolution

Yolanda Schaerli[1,2,*] iD, Alba Jiménez[3], José M Duarte[2], Ljiljana Mihajlovic[1,2], Julien Renggli[4], Mark Isalan[5,6], James Sharpe[3,7,8] & Andreas Wagner[2,9,10,**] iD

## Abstract

**Phenotypic variation is the raw material of adaptive Darwinian evolution. The phenotypic variation found in organismal development is biased towards certain phenotypes, but the molecular mechanisms behind such biases are still poorly understood. Gene regulatory networks have been proposed as one cause of constrained phenotypic variation. However, most pertinent evidence is theoretical rather than experimental. Here, we study evolutionary biases in two synthetic gene regulatory circuits expressed in *Escherichia coli* that produce a gene expression stripe —a pivotal pattern in embryonic development. The two parental circuits produce the same phenotype, but create it through different regulatory mechanisms. We show that mutations cause distinct novel phenotypes in the two networks and use a combination of experimental measurements, mathematical modelling and DNA sequencing to understand why mutations bring forth only some but not other novel gene expression phenotypes. Our results reveal that the regulatory mechanisms of networks restrict the possible phenotypic variation upon mutation. Consequently, seemingly equivalent networks can indeed be distinct in how they constrain the outcome of further evolution.**

**Keywords** constrained evolution; epistasis; gene regulatory networks; regulatory mechanisms; synthetic circuits

**Subject Categories** Development & Differentiation; Evolution; Synthetic Biology & Biotechnology

**Mol Syst Biol. (2018) 14: e8102**

## Introduction

The ability of biological systems to bring forth novel and beneficial phenotypes as a consequence of genetic mutations is essential for evolutionary adaptation and innovation. This ability is encapsulated in the concept of evolvability (Kirschner & Gerhart, 1998; Wagner, 2005b). Evolvability can be limited by evolutionary constraints, which are biases or limitations in the production of novel phenotypes (Smith *et al*, 1985). An example of such constraints comes from laboratory selection experiments with butterfly populations for enhanced wing eyespot colours (Allen *et al*, 2008). Selection was able to increase the amount of black or gold colouring in the two eyespots simultaneously, but was unable to do so for the two different colours independently in the two eyespots. Constrained variation can have multiple genetic and developmental causes that can be difficult to disentangle in a complex developing organism (Arnold, 1992; Wagner, 2011). Therefore, few experimental demonstrations of evolutionary constraints exist. What is more, 30 years after this concept rose to prominence (Smith *et al*, 1985), we still do not understand the mechanistic causes of evolutionary constraints.

The instructions for an organism's development are encoded in gene regulatory networks (GRNs)—networks of interacting transcription factors that control gene expression in both time and space (Davidson, 2006). Mutations in the *cis*-regulatory regions of GRNs play an important part in evolutionary adaptation and innovation (Prud'homme *et al*, 2007; Wray, 2007; Payne & Wagner, 2014). Examples include the evolution of the vertebrate spine (Guerreiro *et al*, 2013), of wing pigmentation in butterflies (Beldade & Brakefield, 2002) and of hindwing reduction in flies (Carroll *et al*, 2001). GRNs are thus primary candidates for systems that might lead to the production of constrained variation (Gompel & Carroll, 2003; Sorrells *et al*, 2015). However, no experimental work exists to find out whether GRNs might constrain

1 Department of Fundamental Microbiology, University of Lausanne, Lausanne, Switzerland
2 Department of Evolutionary Biology and Environmental Studies, University of Zurich, Zürich, Switzerland
3 Systems Biology Program, Centre for Genomic Regulation (CRG), Universitat Pompeu Fabra, Barcelona, Spain
4 Independent Researcher, St-Sulpice, Switzerland
5 Department of Life Sciences, Imperial College London, London, UK
6 Imperial College Centre for Synthetic Biology, Imperial College London, London, UK
7 Institucio Catalana de Recerca i Estudis Avancats (ICREA), Barcelona, Spain
8 EMBL Barcelona, European Molecular Biology Laboratory, Barcelona, Spain
9 The Swiss Institute of Bioinformatics, Lausanne, Switzerland
10 The Santa Fe Institute, Santa Fe, NM, USA
*Corresponding author. Tel: +41 (0) 21 692 56 02; E-mail: yolanda.schaerli@unil.ch
*Corresponding author. Tel: +41 (0) 44 635 61 41; E-mail: andreas.wagner@ieu.uzh.ch

novel gene expression phenotypes, and what the mechanistic causes of such constraints might be. These questions require us to study the relationship between genotypic and phenotypic changes in GRNs. Computational models of gene regulation provide one avenue to understand such genotype–phenotype maps (MacCarthy *et al*, 2003; Wagner, 2005a; Ma *et al*, 2006; Ciliberti *et al*, 2007a,b; Francois *et al*, 2007; Cotterell & Sharpe, 2010; Francois, 2014; Payne & Wagner, 2015). Such models predict that GRNs with different topologies—qualitatively different patterns of interaction between a GRN's genes—can achieve the same gene expression phenotypes, while they differ in their ability to bring forth novel phenotypes through DNA mutations (MacCarthy *et al*, 2003; Ciliberti *et al*, 2007a,b; Francois *et al*, 2007; Jimenez *et al*, 2015; Payne & Wagner, 2015). However, experimental validation of the latter prediction is still lacking.

To help fill the gaps in experimental evidence, we here use the toolbox of synthetic biology. It allows us to create novel GRNs by assembling well-characterised parts. We are therefore no longer limited to studying GRNs *in situ*, that is, in one or few well-studied organisms where influences of genetic background or environment may be difficult to control. Instead, we can construct and modify synthetic GRNs to understand the properties and potential of GRNs to create novel phenotypes (Wall *et al*, 2004; Mukherji & van Oudenaarden, 2009; Lim *et al*, 2013; Wang *et al*, 2016; Bodi *et al*, 2017; Davies, 2017). We previously built multiple 3-gene synthetic networks that display the same gene expression phenotype, but create this phenotype through different regulatory mechanisms (Schaerli *et al*, 2014), where different regulatory dynamics and regulatory interactions among network genes result in different spatiotemporal gene expression profiles (Cotterell & Sharpe, 2010; Schaerli *et al*, 2014; Jimenez *et al*, 2015). The final phenotype is a "stripe" of gene expression (low–high–low) along a spatial axis in response to a chemical concentration gradient that is analogous to a morphogen gradient in development. A GRN's ability to "interpret" a gradient by producing such stripes is crucial in the development of many organisms and body structures, such as axial patterning of the *Drosophila* embryo and vertebrate neural tube differentiation (Stanojevic *et al*, 1991; Wolpert, 1996; Lander, 2007; Rogers & Schier, 2011; Sagner & Briscoe, 2017). The question of which regulatory mechanisms can produce stripes is therefore itself crucial for developmental genetics (Francois *et al*, 2007; Cotterell & Sharpe, 2010). Here, we go beyond this question to ask whether different GRNs that have the same phenotype (a "stripe" of gene expression) can produce different novel (i.e. "non-stripe") gene expression phenotypes in response to mutations, and if so, why.

Specifically, we use here two synthetic circuits that employ different regulatory mechanisms to produce a striped gene expression pattern. Both of these circuits are hosted by *Escherichia coli* bacteria. When these bacteria are grown as a lawn in the presence of a concentration gradient of the morphogen analogue, they display a spatially striped gene expression pattern (Fig 1C). We introduced random mutations into the regulatory regions of these circuits and analysed the resulting phenotypes. The two circuits indeed produce a different spectrum of novel gene expression phenotypes. That is, the gene expression variation they produce is constrained. To identify the mechanistic causes of these constraints, we combined experimental DNA sequence and phenotypic data with a mathematical model of gene expression dynamics.

# Results

## Two networks with distinct regulatory mechanisms differ in their mutant phenotype distributions

Figure 1 shows the topologies (Fig 1A) and the molecular implementations (Fig 1B) of our two starting networks, which we had constructed and characterised previously (Schaerli *et al*, 2014). Briefly, their regulatory input is the sugar arabinose, which serves as a molecular analogue of a developmental morphogen. The arabinose is sensed by the arabinose-responsive promoter pBAD that acts in a concentration-dependent manner. The observable network output is fluorescence, which is produced by superfolder green fluorescent protein (GFP; Pedelacq *et al*, 2006). Positive regulatory interactions are encoded by T7 and SP6 phage RNA polymerases (RNAPs), which start transcription at T7 or SP6 promoters, respectively. Negative interactions are encoded by the transcriptional repressors LacI (lactose operon repressor protein) and TetR (tetracycline repressor). They inhibit transcription when bound to their operator sites (LacO, TetO), which are placed downstream of promoters. The two networks employ distinct mechanisms to produce a gene expression stripe pattern (Cotterell & Sharpe, 2010; Schaerli *et al*, 2014; Jimenez *et al*, 2015). We call these mechanisms the "opposing gradients" and the "concurring gradients" mechanisms. They essentially correspond to the well-studied type 2 and type 3 incoherent feedforward motifs (FFM; Mangan & Alon, 2003; see Box 1 for explanations). Figure 1C schematically shows the temporal expression profiles of the three genes and their steady-state profiles (last panel) of the three genes (colour-coded as in Fig 1A) under varying arabinose concentrations, as previously determined experimentally (Schaerli *et al*, 2014). Whereas the opposing gradients mechanism is known to be involved in *Drosophila melanogaster* anterior–posterior patterning (hunchback, knirps, krüppel; Jaeger, 2011), to the best of our knowledge the concurring gradients mechanism has so far not been observed in a natural stripe-forming regulatory network. However, previous studies added this network to the repertoire of possible stripe-forming mechanisms (Rodrigo & Elena, 2011; Munteanu *et al*, 2014; Schaerli *et al*, 2014).

We introduced mutations into the regulatory regions of these two networks by replacing the wild-type regulatory sequence with semi-randomised weighted oligonucleotides (Isalan, 2006). Resulting average mutation rates per regulatory regions ranged from 2.6 to 3.5 mutations (mainly point mutations and < 5% of insertions and deletions) per regulatory region with individual mutants carrying 1–9 mutations (Dataset EV1, Appendix Table S4). For each of our two networks, we first generated three libraries of mutant networks in which mutations were restricted to regulatory regions of the "red", "blue" or "green" gene (Fig 1). After plating cells from a population whose members harboured a synthetic network variant, we randomly picked colonies, grew them in liquid culture and measured their GFP expression at low (0%), middle (0.0002%) and high (0.2%) arabinose concentrations (Appendix Fig S1, Dataset EV2). We classified the observed fluorescence phenotypes into six categories (Fig 2A; see Materials and Methods for exact definitions): "stripe", "increase", "decrease", "flat" and "broken" (all expression values below a threshold) and "other" (phenotypes that do not fall in any of the previous categories).

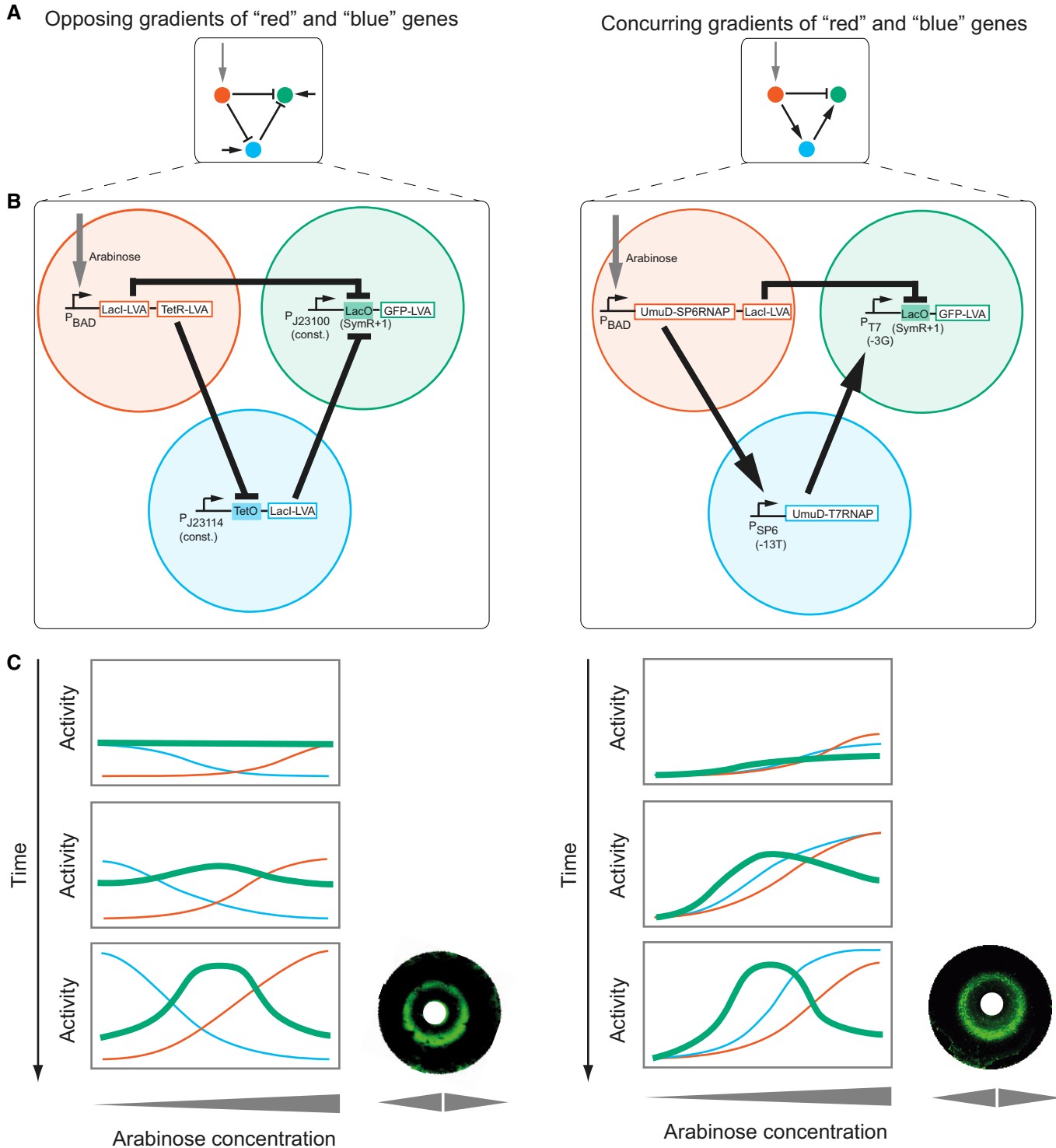

**Figure 1.  Topologies, synthetic implementations and expression profiles of the networks studied.**

A   Topologies of the networks using the *opposing gradients* (left) and *concurring gradients* (right) mechanisms. Arrow: activation; small horizontal arrow: constitutive promoter; bar: repression; red: morphogen input receiver gene; blue: intermediate loop gene; green: stripe output gene.

B   Synthetic implementations of the circuits (Schaerli *et al*, 2014). Open rectangle: open reading frame; filled rectangle: operator; bent arrow: promoter. All genes carry a degradation tag [LVA (Andersen *et al*, 1998) or UmuD (Gonzalez *et al*, 1998)]. Indicated variants of T7 promoter, SP6 promoter and LacO were used (Schaerli *et al*, 2014). J23114 and J23100 are constitutive promoters (http://partsregistry.org/Promoters/Catalog/Anderson).

C   Rectangles: Schematic drawings of spatiotemporal course of gene expression (colour-coded) as in (A) for the two networks (see Box 1). The expression level of the "green" gene is the phenotypic "output" of the network. Corresponding simulations (Code EV1) are shown in Appendix Fig S7. Circles: bacterial lawns display green fluorescent rings as a function of radial arabinose gradients from central paper discs (white). Images were taken 6 h after addition of arabinose. Figure adapted from Schaerli *et al* (2014).

> **Box 1. Two starting circuits producing stripes through two different mechanisms**
>
> *Opposing gradients mechanism (Incoherent FFM type 2):* The "red" gene [with the open reading frames (ORFs) for LacI and TetR encoded on the same transcript] is activated by the "morphogen" arabinose (vertical arrow). Its products thus form a gradient of increasing concentration with increasing arabinose concentration. The "blue" gene (LacI) and the "green" gene (GFP) are expressed from constitutive promoters. However, the "blue" gene is also repressed by the "red" gene product (TetR). Thus, the "blue" gene product forms an *opposing* gradient with respect to the gradient of the "red" gene product. Both the "blue" (LacI) and "red" (LacI) gene products repress the "green" gene. The GFP thus reaches a high expression only at medium morphogen concentration where the repression from the "red" and "blue" genes is low.
>
> *Concurring gradients mechanism (Incoherent FFM type 3):* The "red" gene (with the ORFs for SP6 RNA polymerase (RNAP) and LacI encoded on the same transcript] is activated by the "morphogen" arabinose, just as in the previous circuit. Its expression thus also mimics the arabinose gradient. However, in this circuit the "red" gene product SP6 RNAP activates the "blue" gene, which thus forms a *concurring* gradient with respect to the gradient of the "red" gene product. The "green" gene is activated by the "blue" gene (T7 RNAP) and repressed by LacI of the "red" gene. Its maximum expression occurs at medium arabinose concentration where there is already activation from the "blue" gene, but not yet a high level of repression of the "red" gene.

Figure 2B summarises the spectrum of phenotypes we observed after mutagenesis. We first note that both networks are to some extent robust to mutations; that is, a considerable fraction of mutations do not change the "stripe" phenotype (black sectors in Fig 2B). What is more, the two types of networks we study differ in their robustness. Averaged across the three genes, 45.5% of analysed mutants preserve the "stripe" phenotype in the concurring gradients network, whereas only 32.9% do so in the opposing gradients network. The concurring gradients network is thus significantly more robust to mutations [Chi-square goodness-of-fit test, $\chi^2$ $(1, N = 215) = 13.67$, $P = 0.0002$]. Next, we note that within any one of the two networks the *novel* phenotypes do not occur at the same frequency, providing evidence for the biased production of novel phenotypes, where certain types of phenotypes are more common than others.

We also observed differences in the types of novel phenotypes between the two networks. For example, 8.2% of mutants of the opposing gradients networks show a "flat" GFP expression phenotype, where the GFP expression is invariant to arabinose concentrations (yellow sector in Fig 2B). In contrast, mutations in the concurring gradients network did not produce a single such phenotype. In addition, mutations in the opposing gradients network are more likely to create a "decrease" phenotype (purple, 29.8% of all novel phenotypes) rather than an "increase" phenotype (orange, 15.4%). For the concurring gradients network, the opposite is true: mutations are more likely to create "increase" (23.0%) rather than "decrease" (18.1%) phenotypes.

Next, we analysed the GFP expression levels of the measured phenotypes quantitatively (Fig 2C). To this end, we compared the GFP expression at medium arabinose concentration to those at high (*y*-axis) and at low arabinose concentrations (*x*-axis). We note that

the previously classified phenotypes (Fig 2A) form well-separated clusters in this analysis. For example, networks in the bottom-right quadrant correspond to "stripe" phenotypes, because their pattern is described as an increase (positive *x*-axis) followed by a decrease (negative *y*-axis) in expression. Consequently, "decrease" and "increase" phenotypes occupy the upper-right and bottom-left quadrants, respectively. We also sequenced the mutated regulatory regions of all analysed networks and find a weak association between the number of mutations a network carries, and the extent to which its observed phenotype differs from the starting "stripe" phenotype (as quantified through the Euclidean distance; Appendix Fig S2).

Subsequently, we analysed the differences in novel phenotypes created by mutations in specific regulatory regions (i.e. of the "red", "blue" or "green" gene). Within any one of the two network types, regulatory mutations in the "red" gene most often create "increase" phenotypes (Fig 2D, pie charts left to the "red" genes), whereas those in the "blue" gene most often create "decrease" phenotypes (Fig 2D, pie charts at the bottom of the "blue" genes), and those in the "green" gene preferably create "broken" phenotypes (Fig 2D, pie charts to the right of the "green" genes). As a consequence, not all phenotypes can be reached by introducing mutations in the regulatory region of any of the three genes. For example, in the opposing gradient network, the "increase" phenotype is only reachable by introducing mutations into the "red" gene, but not in the "blue" and "green" genes.

The two networks differ in the spectrum of novel phenotypes that mutations in individual genes create, which is especially obvious for mutations in the "green" gene: unless regulatory mutations in this gene lead to a complete loss of expression ("broken"), the opposing gradients network is > 5 times more likely to create a "flat" phenotype (23.2%) than a "decrease" phenotype (4.1%). In contrast, the concurrent gradients network does not produce any "flat" phenotype at all, but readily produces "increase" phenotypes (4.5%). In sum, mutations in networks which start with the same phenotype (single "stripe" formation), but which have alternative topologies and regulatory mechanisms, create different kinds of novel phenotypes. Hence, phenotypic variation is subject to constraints, and these constraints differ between regulatory regions and networks.

## Differences in constrained variation can be explained by differences in the regulatory mechanisms behind stripe formation

We next asked whether the regulatory mechanisms contributing to stripe formation can help explain these phenotypic constraints. In doing so, we focused on novel phenotypes produced by regulatory mutations in the "green" gene, because such mutations produced the most distinct spectrum of novel phenotypes (Fig 2D). Also, the regulation of this gene is most complex, because it receives two regulatory inputs instead of just one for the other genes (Fig 1). (Similar analyses for the "red" and "blue" genes can be found in Appendix Figs S3–S5.)

To address this question, we first used a mathematical model that we had developed previously and validated experimentally to describe the regulatory dynamics of our networks (Schaerli *et al*, 2014). Briefly, the model uses Hill-like functions to represent gene

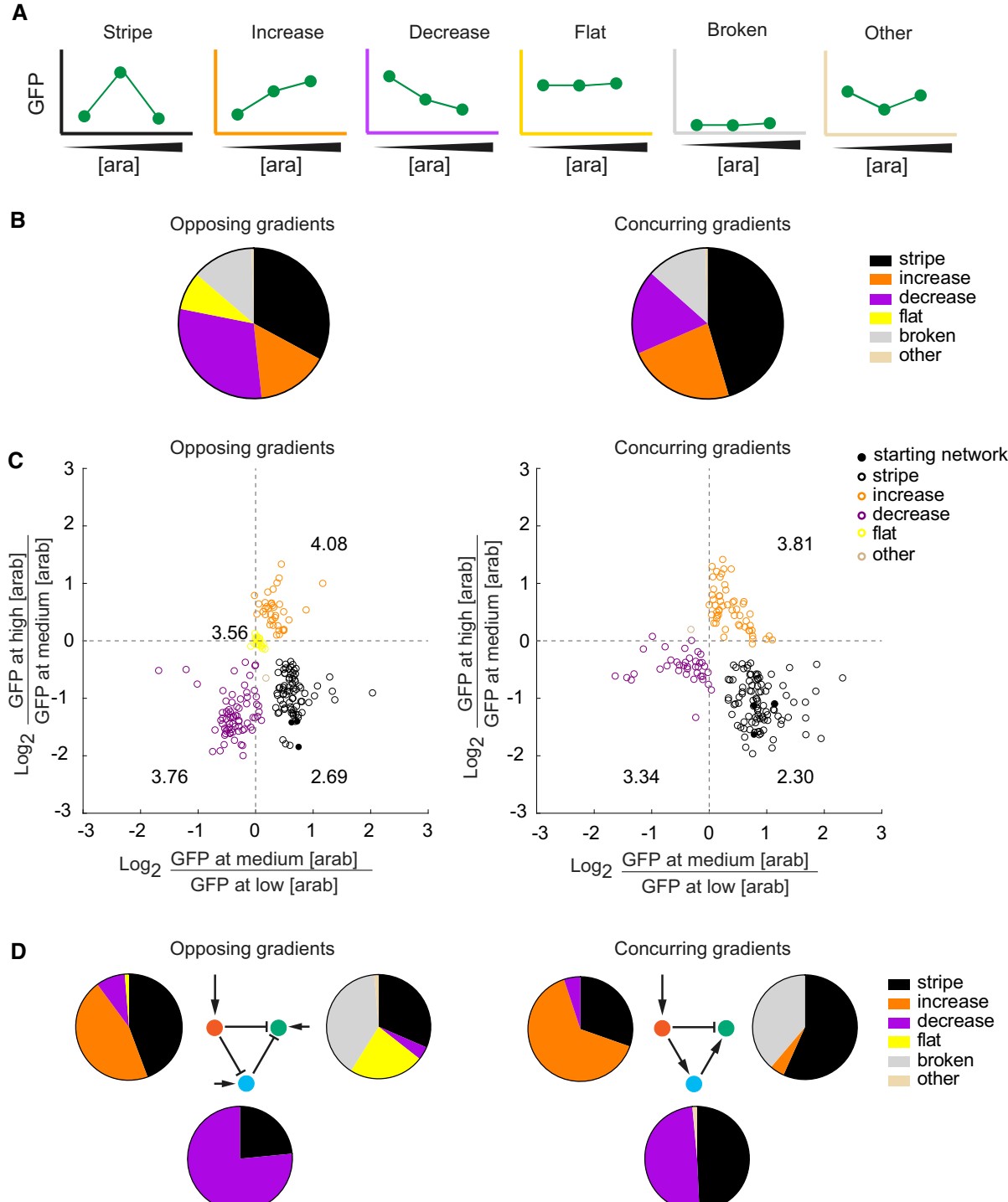

**Figure 2.   Different networks create different spectra of novel phenotypes after mutation.**

A   Phenotype categories used in this study. See Materials and Methods for exact definitions. [ara], arabinose concentration. The colours of the axes are used throughout the paper to colour-code the phenotypes.

B   Experimentally observed phenotype distributions when mutating one regulatory region at a time for the opposing (left) and concurring (right) gradients networks. The pie charts summarise the spectrum of all mutant phenotypes observed in a network. The data are based on 234 and 215 mutants of the opposing and concurring gradients networks, respectively.

C   The GFP expression level (fluorescence normalised by the absorbance) of each individual mutant at medium arabinose concentration is compared to the GFP expression levels at low (*x*-axis) and high arabinose (*y*-axis) concentrations. The numbers written close to each phenotype group are the average mutation rates for that group. We omitted the "broken" phenotype from this analysis, as the networks with this phenotype do not show any significant GFP expression.

D   Experimentally observed phenotype distributions as displayed in (B and C), grouped according to the mutated gene.

regulation changes based on equilibrium binding of transcription factors to their DNA binding sites (Bintu *et al*, 2005; see Table 1, and Appendix Model Description and Appendix Tables S1 and S2 for details). The unmutated ("wild-type") model for each circuit used parameter values determined in our previous study (Schaerli *et al*, 2014). Into these models, we now introduced quantitative changes in the parameters relating to the promoter activity (binding constants of activators and transcription rates) and to the operator activity (binding constants of repressors), in order to predict phenotypes that are accessible by mutations (see Materials and Methods for details). We represent the unmutated network as a point in parameter space, and study regions near this point that are accessible by mutations, and the novel phenotypes they contain (Dichtel-Danjoy & Felix, 2004). For each parameter we varied, we chose to examine a uniform distribution in a range between zero and 110% of the starting/wild-type parameter values, because available mutagenesis data for the components used in the "green" genes of our synthetic circuits (Niland *et al*, 1996; Imburgio *et al*, 2000; Shin *et al*, 2000) suggest that most mutations decrease a parameter value rather than increasing it.

We visualise the results in phenotype diagrams (Fig 3A and Appendix Fig S2), which are projections of the higher-dimensional parameter space onto two dimensions (Jimenez *et al*, 2015). These diagrams are built as pixelated images in which for each combination of parameter values (for each "pixel") the model predicts the resulting phenotype and assigns the corresponding colour (see legend, Fig 3A). In these diagrams, a parameter value of 100% corresponds to the wild-type value and other values in this region are expressed as a percentage of the wild-type. For example, the black region in Fig 3 corresponds to mutant parameter combinations that maintain the "stripe" phenotype. Its area is therefore a measure for a network's robustness to parameter changes. Overall, these diagrams provide information on which parameters must be mutated, and by how much, in order to access a given phenotype.

For each of the novel phenotypes observed experimentally when mutating the regulatory region of the "green" gene (Fig 2B), some "mutated" model parameter values exist that reproduce the phenotype (Fig 3A). The phenotype diagrams of the two networks (Fig 3A) are visually very distinct, indicating that the two networks differ in their potential to access specific phenotypes. Specifically, for the opposing gradients network we find regions corresponding to the "broken" (grey), "decrease" (purple), "flat" (yellow) and "other" (beige) phenotypes, whereas for the concurring gradients network we find regions for the "broken" and "increase" (orange) phenotypes—corresponding to the phenotypes observed experimentally when mutating the "green" gene (Fig 2D). Especially instructive are mutants with strongly decreased repressor binding (i.e. reduced operator activity, arrows in Fig 3A). Such mutants produce a "flat" phenotype (yellow region) in the opposing gradient network, but an "increase" phenotype (orange region) in the concurring gradient network (Fig 3A). In other words, even though both networks contain the same operator (LacO) in the "green" gene, the model predicts that identical operator mutations can lead to different novel phenotypes. Figure 3B illustrates how this is possible: if an operator mutation removes the incoming negative interaction of the "green" gene in the opposing gradient network, the constitutive promoter becomes the sole driver of "green" gene expression. Consequently, GFP expression becomes independent of arabinose concentrations, which results in a "flat" phenotype. In contrast, after removing the repression of the "green" gene in the concurring gradient network, the "green" gene is still regulated by the activating "blue" gene (T7 RNAP) in an arabinose-dependent manner. Hence, in this mutant circuit, GFP expression increases with increasing arabinose concentrations. In sum, different biases in the production of novel phenotypes can be explained by differences in regulatory mechanisms.

**Sequence analysis confirms phenotype diagram predictions of constrained phenotypic variation**

We next validated the predictions made by our phenotype diagrams with DNA sequence analysis. To this end, we analysed the sequences of the regulatory regions of the "green" genes we had mutagenised. Because many mutagenised circuits have multiple regulatory mutations, we first categorised circuits according to the number of mutations that they contained, and studied the frequency of observed phenotypes in each category (Fig 4A, large diagrams: all mutations). We followed the same procedure for the subsets of circuits that have mutations only in the promoter sequence or only in the operator sequence (Fig 4A, smaller diagrams). This categorisation reveals that mutations in the operator can produce a "flat" phenotype in the opposing gradient network, but an "increase"

**Table 1. Model (Schaerli *et al*, 2014) and biological meaning of parameters for the "green" genes of the opposing and concurring gradients networks, respectively.**

| Definition | Name | Parameter relates to |
|---|---|---|
| Opposing gradients | *a* | Basal transcription rates from the free promoter |
| | *b* | Transcription rate when LacI is bound |
| $GFP = \frac{a + b(c\,LacI)^n}{1 + (c\,LacI)^n}$ | *c* | Binding constant of LacI |
| | *n* | Hill coefficient (multimerisation or cooperativity) |
| Concurring gradients | *a* | Basal transcription rate in absence of T7 RNAP |
| | *b* | Transcription rate when T7 RNAP is bound |
| | *c* | Binding constant of T7 RNAP |
| | *d* | Binding constant of LacI |
| $GFP = \frac{a + b(c\,T7)^n + ef(c\,T7)^n(d\,LacI)^m}{1 + (c\,T7)^n + (d\,LacI)^m + f(c\,T7)^n(d\,LacI)^m}$ | *e* | Transcription rate when T7 RNAP + LacI are bound |
| | *f* | Cooperativity/competition constant of T7 RNAP/LacI |
| | *n* | Hill coefficient (multimerisation or cooperativity) |
| | *m* | Hill coefficient (multimerisation or cooperativity) |

The complete model for both networks can be found in the Appendix Tables S1 and S2.

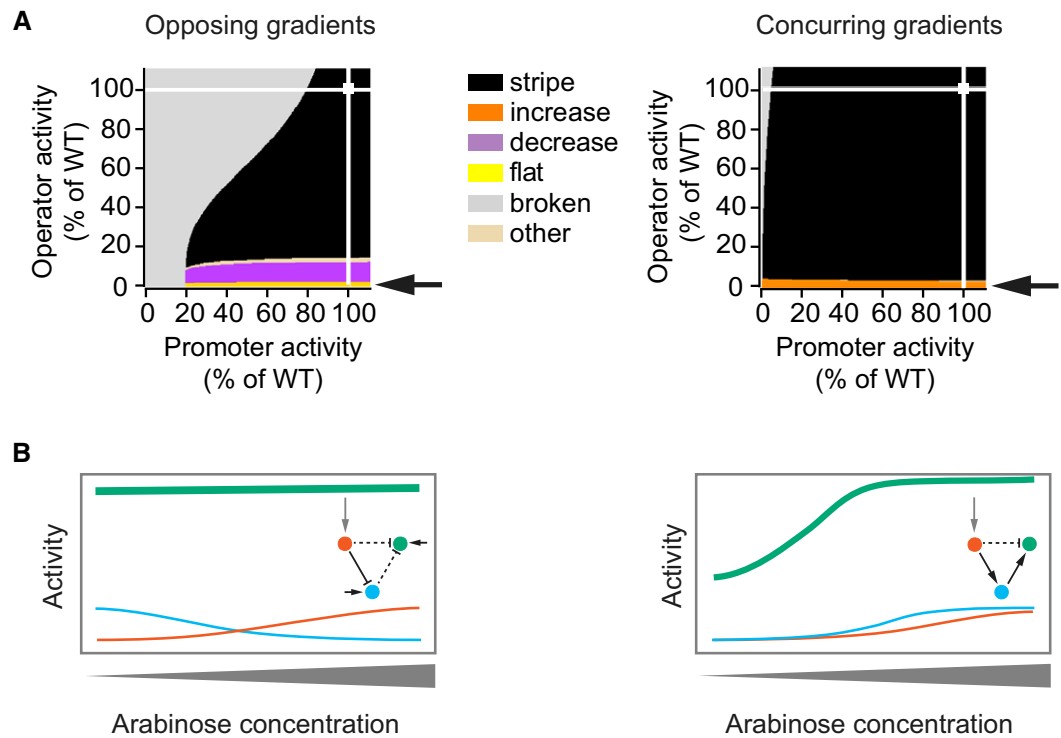

**Figure 3.  The same parameter change leads to different phenotypes in the two network types.**

A   Phenotype diagrams for parameters that describe the activity of the "green" gene. Horizontal and vertical axes indicate promoter and operator activities of the "green" gene relative to the wild-type value (WT, 100%). All parameters affecting the promoter or operator were varied jointly and to the same extent. Colours indicate phenotypes predicted by the model over the whole range of promoter and operator activity values. White squares indicate the parameter combination of the unmutated circuit, which produces the "stripe" phenotype, and white lines are visual guides that project these values onto the two parameter axes. Arrows point to the phenotype observed when operator activity decreases to a value near zero per cent.

B   Schematic drawing of how a strong reduction in operator activity of the "green" gene affects gene expression patterns differently in the two types of networks. Insets: topologies of the networks, with dashed lines indicating interactions affected by mutations in the operator.

phenotype in the concurring gradient network (Fig 4A, smaller diagrams), thus validating model predictions (Fig 3). In addition, the model predicts that mutations in the operator of the opposing gradient network are able to produce a "decrease" phenotype (Fig 3A). Even though we did not find a circuit with a "decrease" phenotype that has *only* operator mutations, all circuits with this phenotype carry at least one mutation in the operator (and additional mutations in the promoter, Dataset EV1).

Subsequently, we analysed the locations of mutations in networks with a given phenotype in greater detail (Fig 4B for mutations in the "green" gene, Appendix Fig S5 for mutations in the other genes). Many networks carry mutations in both the promoter and the operator. Nevertheless, for the "flat" phenotype of the concurring gradients network, operator mutations are significantly enriched [Chi-square goodness-of-fit test, $\chi^2$ (1, $N = 59$) = 7.64, $P = 0.006$], as predicted by the model (Fig 3). For the "decrease" phenotype in the opposing gradients network and the "increase" phenotype in the concurring gradients network, our dataset is too small to detect the predicted enrichment of operator mutations.

Especially informative are those mutants with a novel phenotype that carry only a single point mutation (red arrows in Fig 4B). Among them are two different mutants of the opposing gradient network with a "flat" phenotype. The mutations in them affect the

two central nucleotides of the Lac operator, which are known to be critical for operator function (Lehming *et al*, 1987; Zhang & Gottlieb, 1995; Falcon & Matthews, 2000; Kalodimos *et al*, 2001). Mutations in these positions reduce the operator's binding affinity for the LacI repressor dramatically (Lehming *et al*, 1987), and the observed "flat" phenotype for these mutants supports our phenotype diagram predictions.

**Regulatory mechanisms influence the phenotype distributions more than the actual parameters of the network**

So far, we demonstrated that each of the two analysed networks yields a biased spectrum of novel phenotypes after introducing mutations, and that two networks with different regulatory mechanisms yield different spectra of novel phenotypes. However, these spectra may not be influenced only by a network's regulatory mechanisms. They may also differ among networks with the same topology and the same regulatory mechanism, but with quantitative differences in the biochemical parameters that determine a networks gene expression pattern. To find out whether this is the case, we performed the following experiments: we took two mutant stripe-forming networks of the concurring gradient mechanism with mutations in all three genes (mutants A and B) and

introduced further mutations into their "green" regulatory regions. Figure 5A shows the resulting phenotype distributions and compares them to the initial ("wild-type", WT) network. As in the WT network, we observe "stripe", "broken" and "increase" phenotypes in the mutants. However, the figure also shows that the proportions of these phenotypes differ among the networks. In addition, 3% (mutant A) and 1.3% (mutant B) of the two concurring gradient network variants displayed a "decrease" phenotype. This suggests that by making "neutral" or "silent" genetic changes in a regulatory network that do not affect its ("stripe") phenotype, new phenotypes can become accessible through further mutations (Schuster *et al*, 1994; Dichtel-Danjoy & Felix, 2004; Wagner, 2011). Nevertheless, the phenotype distributions observed for the three networks with the concurring gradients mechanism are more similar to each other than to the one of the opposing gradients network (Fig 5B). For example, we did not observe any "flat" phenotype—a phenotype very frequently produced by mutations in the opposing gradients network. In sum, based on these experiments, the evolution of new phenotypes in our study networks is

more constrained by the regulatory mechanism itself than by the actual network parameters.

## Phenotype distributions can be explained by the model

Encouraged by the agreement between phenotype diagrams and mutational data, we also aimed to see whether a model of mutational effects can correctly fit the *frequencies* instead of just the *kinds* of phenotypes caused by mutations. To find out, we simulated the effects of mutations by changing specific parameters of the model. If a mutation affected a gene's promoter (or operator), we changed all the parameters determining promoter (or operator) activity. Some parameters were changed to the same extent (i.e. we set the parameters to the same percentage of their wild-type parameter value), because a mutation is likely to affect these specific parameters in a similar way (Appendix Table S3). We drew the changed parameters from a uniform distribution, and for each parameter, we aimed to identify upper and lower bounds for this distribution that give the best possible agreement between the

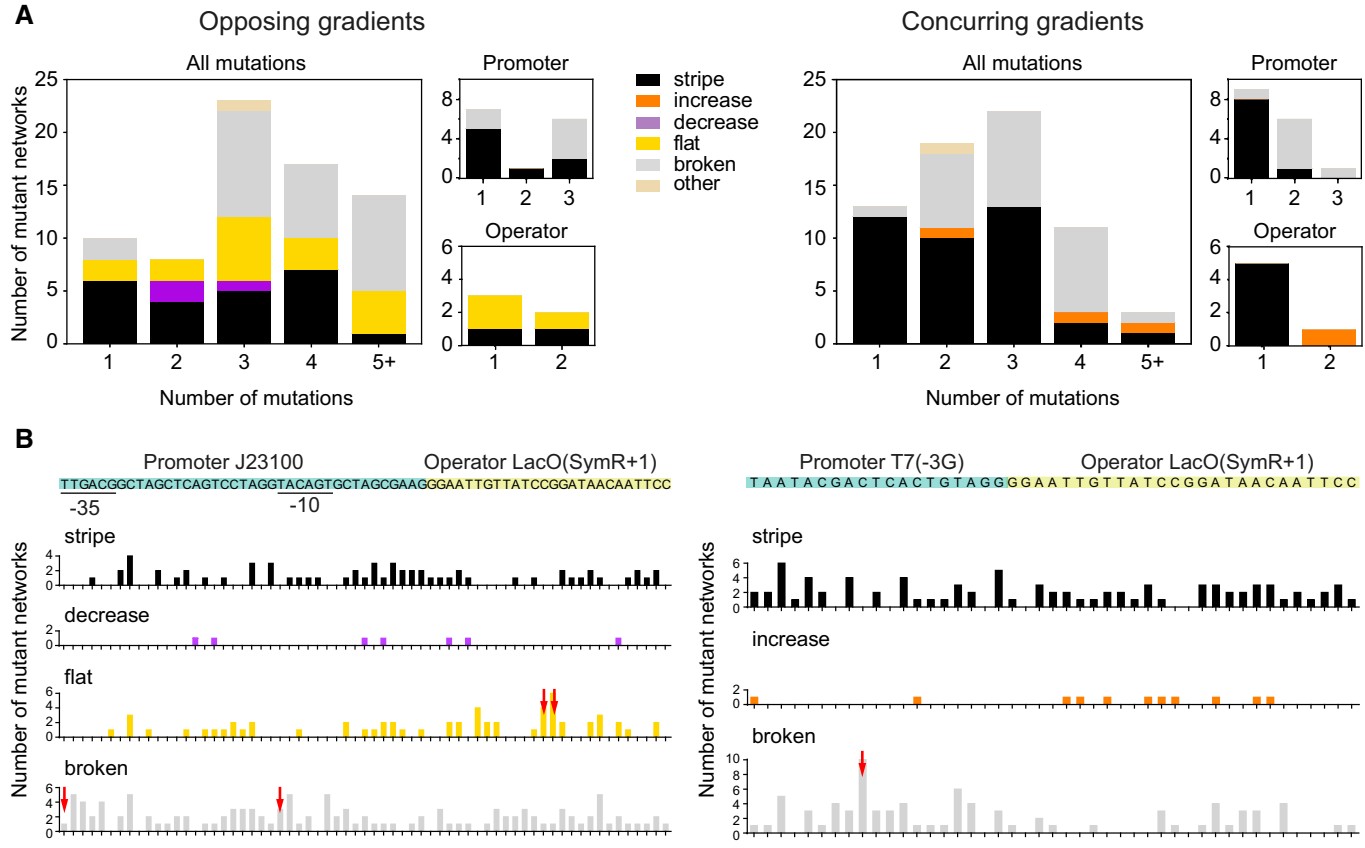

**Figure 4. Sequence analysis of the regulatory regions.**

A   Distribution of observed phenotypes of mutant circuits with mutations in the regulatory region of the "green" gene. Phenotypes are colour-coded (legend). Smaller graphs to the right indicate the subset of networks that have mutations only in the promoter or only in the operator. The data are based on 73 and 67 mutants of the opposing and concurring gradients networks, respectively.
B   Wild-type sequences of regulatory regions (top of each panel, important elements labelled and coloured) together with the number of mutations at each site of a regulatory region that produce phenotypes of a given kind (bar-charts below sequence, phenotypes labelled and colour-coded). The height of each bar corresponds to the number of mutant networks with a mutation at a given position, where these mutations produced the indicated phenotype. Only phenotypes produced by at least three mutant circuits are shown. Red arrows indicate genotypes that can produce a novel phenotype with a single mutation at the indicated position.

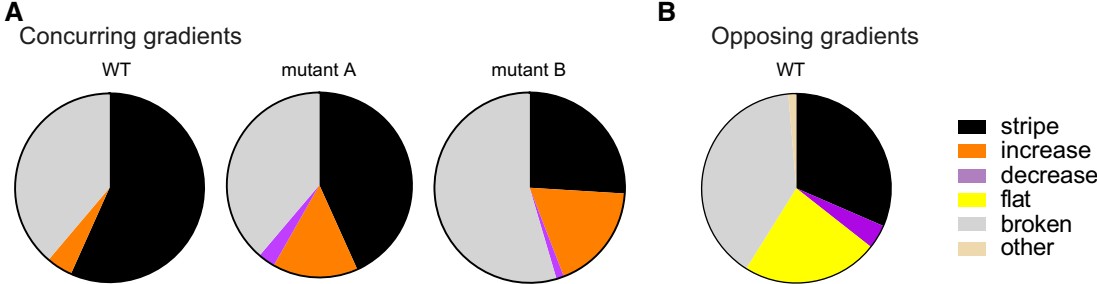

**Figure 5.  Phenotypes are more constrained by the regulatory mechanism itself than by the actual network parameters.**

A   Mutations were introduced into the "green" genes of three concurring gradients networks with different biochemical parameters. The pie charts display the observed phenotype distributions. The data are based on 68 (WT, Fig 2D), 67 (mutant A) and 77 (mutant B) variants.

B   For comparison, we show again the phenotype distribution of the "green" gene of the opposing gradients network (Fig 2D).

experimental and model data (Appendix Table S3). Guided by the phenotype diagrams, we were indeed able to identify such bounds, which enabled our model to reproduce experimental phenotype distributions for all three genes well (Table 2), and in particular for the "blue" and "green" genes where Chi-square tests indicate no significant difference ($P > 0.05$) between the experimental and model phenotype distributions.

For 20 out of 25 parameters, the lower and upper bounds of the identified intervals were equal or below the WT parameter value (=100%), consistent with the notion that most random mutations will disrupt molecular interactions and thus decrease the corresponding parameter's value, which is also in agreement with available mutagenesis data of the components used in our synthetic circuits (Niland *et al*, 1996; Imburgio *et al*, 2000; Shin *et al*, 2000). One exception were the parameter values describing the basal transcription promoter activity ("leakiness") of the (pBAD) promoter in the "red" genes of both networks (opposing gradients: $a_T$, $a_L$, concurring gradients: $a_L$, $a_S$). These values have upper bounds higher than 100% of the unmutated (WT) value (Appendix Table S3), which is consistent with the DNA looping mechanism of the pBAD promoter (Lobell & Schleif, 1990; see Appendix Discussion of Lower and Upper Bounds of the Parameter Intervals for details). Another exception was the basal transcription promoter activity ("leakiness") of the SP6 promoter in the concurring gradient network ($a_T$). It also has an

upper bound higher than 100% of the unmutated (WT) value (Appendix Table S3). This was unexpected and led us to discover a context-dependent effect in the plasmid we used to express the synthetic circuits (see Appendix Discussion of Lower and Upper Bounds of the Parameter Intervals for details). In sum, we were able to reproduce the experimental phenotype distributions with our model by identifying ranges in which "mutated" parameters fall. Moreover, these ranges are in agreement with known mutational effects.

The most significant difference between the phenotype distributions of the experiments and the model is the fraction of networks displaying phenotypes in the "other" category, which is consistently higher in the model predictions than in our experimental data. This can be explained by the fact that we excluded any network from further analysis that displayed phenotypes falling into different categories in at least one of three replicate phenotype measurements (see Materials and Methods for details). This was often the case for the "other" phenotype, because it is situated in a narrow range of the phenotype diagram between two other phenotype categories. For example, networks show this phenotype if they have lower expression levels at low and high arabinose concentrations than at medium arabinose concentration, but neither satisfy our (stringent) definition of a "stripe", nor that of any of the other phenotype definitions (see, e.g., in Fig 3A for opposing gradient network). Small

**Table 2.  Experimentally observed phenotype distributions for the "red", "blue" and "green" genes can be reproduced by the model.**

| | Opposing gradients | | | | | | Concurring gradients | | | | | |
|---|---|---|---|---|---|---|---|---|---|---|---|---|
| | Red | | Blue | | Green | | Red | | Blue | | Green | |
| | e | m | e | m | e | m | e | m | e | m | e | m |
| Stripe | 44.3 | 48.1 | 23.5 | 27.4 | 31.5 | 27.6 | 30.4 | 14.8 | 49.3 | 45.3 | 56.7 | 54.1 |
| Increase | 45.6 | 24.1 | 0.0 | 0.0 | 0.0 | 0.0 | 64.6 | 75.9 | 0.0 | 0.0 | 4.5 | 1.8 |
| Decrease | 8.9 | 11.5 | 76.5 | 67.5 | 4.1 | 7.2 | 5.1 | 2.6 | 49.3 | 42.9 | 0.0 | 0.0 |
| Flat | 1.3 | 7.5 | 0.0 | 0.0 | 23.3 | 14.8 | 0.0 | 1.7 | 0.0 | 0.0 | 0.0 | 0.0 |
| Broken | 0.0 | 0.0 | 0.0 | 0.0 | 39.7 | 44.0 | 0.0 | 0.0 | 0.0 | 1.1 | 38.8 | 43.7 |
| Other | 0.0 | 8.8 | 0.0 | 5.1 | 1.4 | 6.5 | 0.0 | 5.1 | 1.4 | 10.7 | 0.0 | 0.4 |

e, Experimentally observed phenotype distributions (in %) when mutating one regulatory region at the time (Fig 2D). m, Phenotype distributions produced by the model when mutating one regulatory region at a time.

amounts of variation in replicate phenotype measurements can therefore lead to different phenotype classifications between replicates (and subsequent exclusion of a network from further analysis) in our experiments, but not in the model, which does not incorporate this source of variation.

**Non-additive interactions of mutations in multiple regulatory regions are explained by the regulatory mechanisms of the networks**

Because mutations rarely occur in isolation in a single gene, we next asked whether mutations in different regulatory regions independently affect gene expression phenotypes. To this end, we pooled networks with mutations in single regulatory regions to obtain networks with mutations in the regulatory regions of two or three genes. We then measured the gene expression phenotypes of these multiple-gene mutants (Dataset EV2) and sequenced their regulatory regions (Dataset EV1). Figure 6B shows the resulting distribution of phenotypes. Similar to the one-gene mutants (Fig 2B, repeated in Fig 6A), some phenotypes occur more frequently than others, and the opposing and concurring gradient networks produce any one phenotype at different frequencies.

Regardless of the regulatory mechanism, the frequencies of novel phenotypes differed significantly between networks with mutations in multiple versus single genes (Fig 6A and B; opposing: [Chi-square, $\chi^2$ (4, $N = 36$) = 41.7, $p < 0.0001$]; concurring: [Chi-square, $\chi^2$ (4, $N = 41$) = 167.0, $p < 0.0001$]). For example, both networks produce the "flat" phenotype in response to multiple mutations, but the concurring gradient networks did not produce this phenotype in response to single-gene mutations (compare yellow sectors in Fig 6A and B).

Because the phenotypes we observed in the multiple-gene mutants are not just additive superpositions or "sums" of phenotypes observed when the mutations occur separately, the mutations in the different genes must interact non-additively (epistatically) to produce novel phenotypes, such that a mutation's phenotypic effect depends on the genetic background in which it occurs (Lehner, 2011; Mackay, 2014).

In Appendix Fig S6, we show an experimental example of how mutations in the "green" and "blue" genes can interact to produce a "flat" phenotype in the opposing gradient network: the network with the mutated "green" gene maintains the "stripe" phenotype (Appendix Fig S6A), and the network with the mutated "blue" gene leads to a "decrease" phenotype (Appendix Fig S6B). When these two mutations are combined, the resulting phenotype is "flat" (Appendix Fig S6C). Importantly, this new phenotype cannot just be explained as an additive superposition of the two individual phenotypes.

To understand the phenotype distributions of the multiple-gene mutants, and in particular the non-additive interactions, we turned again to our model. Analogous to the experiments, we now changed parameters of multiple genes simultaneously, within the exact same ranges as used to model single-gene mutants (Appendix Table S3). The resulting phenotype distributions (Fig 6C, Appendix Table S5) predict the experimentally observed distributions (Fig 6B) very well, with a Chi-square test indicating no significant difference between experiment and prediction (opposing: [Chi-square, $\chi^2$ (5, $N = 36$) = 6.962, $P = 0.2235$]; concurring: [Chi-square, $\chi^2$ (5, $N = 41$) = 5.552, $P = 0.3522$]). This implies that both constrained variation and non-additive interactions of mutational effects are a direct consequence of how individual network genes interact with each other.

# Discussion

Mutations in regulatory regions of GRNs play a crucial role in evolutionary adaptation and innovation (Prud'homme *et al*, 2007; Wray, 2007; Payne & Wagner, 2014). Here, we first introduced random mutations in the regulatory regions of two synthetic stripe-forming GRNs (Fig 1) and analysed the resulting distributions of novel gene expression phenotypes (Fig 2). Both networks produced a non-uniform distribution of novel phenotypes and are thus inconsistent with a naïve expectation (null model) that each non-stripe phenotype is produced at the same frequency. More interestingly, the different networks displayed different phenotypic variation and consequently different constraints in the production of novel phenotypes. The identity of the mutated regulatory region and non-additive interactions among mutations in multiple regions also influenced these constraints.

A mathematical model describing the regulatory mechanisms of the two networks allowed us to understand the differences between accessible novel phenotypes for the two networks (Figs 2 and 3). The model predictions are also supported by DNA sequencing data (Fig 4). We thus provide for the first time empirical evidence that GRNs with different regulatory mechanisms can cause different constrained variation, as was recently proposed (Jimenez *et al*, 2015). We also provide experimental evidence that the mechanism by which a network produces a stripe constrains the origin of novel expression phenotypes more than quantitative parameters driving gene expression dynamics (Fig 5).

Comparisons of GRNs in related species indicate that they indeed solve the problem of producing a specific adaptive phenotype in many different ways, and that these solutions diverge substantially on evolutionary time scales, even when the ultimate phenotype stays qualitatively the same (Savageau, 1983; Weiss & Fullerton, 2000; True & Haag, 2001; Dalal & Johnson, 2017; Johnson, 2017). Examples include the GRN that regulates mating in yeast: even though both *Saccharomyces cerevisiae* and *Candida albicans* produce two mating types (a-cells and α-cells), the circuit responsible for determining these mating types has changed substantially during evolution (Tsong *et al*, 2006; Sorrells *et al*, 2015). Why a specific GRN and not one of its alternatives evolves remains an open and important question (Carroll, 2008). In an attempt to understand the pertinent principles of GRN evolution, Savageau formulated its "demand rule" (Savageau, 1977). He observed that activators and repressors can achieve the same regulatory goals, but that frequently expressed genes tend to be regulated by activators (positive mode of regulation), whereas rarely expressed genes tend to be regulated by repressors (negative mode). These differences can be explained by the fact that negative and positive regulatory modes can lead to different phenotypes and to different deleterious consequences upon mutation that favour one or the other mode of regulation (Savageau, 1977, 1983, 1998a,b). While Savageau's work focuses on maintaining the initial regulation, our observations show that seemingly equivalent

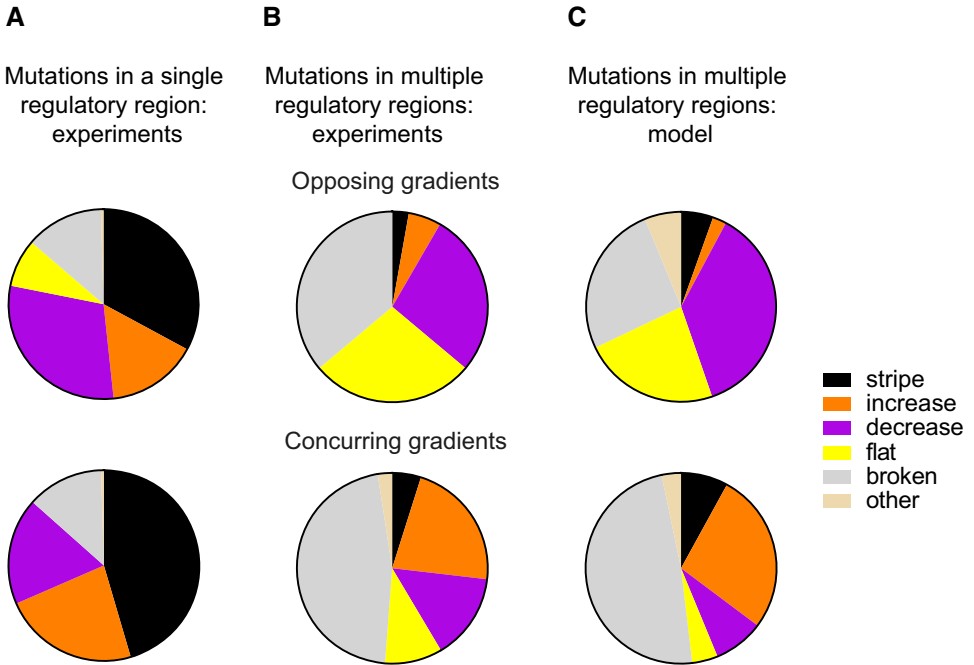

**Figure 6.  Mutations in multiple regulatory regions interact non-additively.**

A  Experimentally observed phenotype distributions when mutating one regulatory region at a time for the opposing (top) and concurring (bottom) gradients networks. Data redisplayed from Fig 2B to facilitate comparison with Fig 6B.
B  Experimentally observed phenotypes of networks with at least two mutated regulatory regions. The data are based on 36 and 41 mutant networks for the opposing (top) and concurring (bottom) gradients mechanisms, respectively.
C  Phenotype distributions produced by the model when simultaneously mutating multiple regulatory regions.

solutions to solve a biological problem also differ in the qualitatively novel phenotypes (which might be adaptive) that they can access through mutation.

While evolutionary constraints are an important concept in evolutionary theory (Smith *et al*, 1985; Arnold, 1992), few experimental studies have aimed to detect or quantify them (Teotonio & Rose, 2000; Beldade *et al*, 2002; Frankino *et al*, 2005; Allen *et al*, 2008; de Vos *et al*, 2013, 2015; Bolstad *et al*, 2015; Lagator *et al*, 2016, 2017a; Zalts & Yanai, 2017). These studies emphasise the importance of natural selection in determining the outcome of adaptive evolution (Beldade *et al*, 2002; Frankino *et al*, 2005), but they also show that evolution of development can be biased by the constrained production of phenotypes (Kiontke *et al*, 2007; Allen *et al*, 2008; Bolstad *et al*, 2015). Laboratory selection experiments have proven to be a powerful tool for detecting evolutionary constraints (Kiontke *et al*, 2007; Allen *et al*, 2008; Bolstad *et al*, 2015), for example in the colour pattern of butterfly wing spots (Allen *et al*, 2008). However, they have been less successful in disentangling their genetic and developmental causes (Arnold, 1992; Wagner, 2011). This is not surprising, because studying the evolution of developmental GRNs in a multicellular organism is extremely difficult: GRNs are complex, highly interconnected, often incompletely understood, and their genes may be highly pleiotropic, serving multiple functions outside any one GRN (Stearns, 2010). In addition, essential molecular tools for manipulating GRNs are often unavailable, especially in non-model organisms.

Experimental studies on proteins (Miller *et al*, 2006; Harms & Thornton, 2013) and *cis*-regulatory elements (de Vos *et al*, 2013, 2015; Lagator *et al*, 2016, 2017a) with simple phenotypes have demonstrated how structure–function relationships of macromolecules can constrain evolution. Here, we extended this approach to nonlinear GRNs by studying synthetic circuits in *E. coli* (Fig 1). We performed experiments with easily modifiable, well-characterised synthetic circuits that are not essential for the survival of their host organism. However, the chosen phenotype—stripe formation in a chemical gradient—is crucial for embryonic development of many organisms and body structures (Stanojevic *et al*, 1991; Wolpert, 1996; Lander, 2007; Rogers & Schier, 2011; Sagner & Briscoe, 2017). For example, an opposing gradients network is part of the gap network responsible for axial patterning in the *Drosophila* embryo (Jaeger, 2011). And while a lawn of *E. coli* cells carrying a synthetic GRN does not capture the complexity of a developing animal, this reduced complexity also allowed us to study the potential of GRNs to bias phenotype production without confounding effects. In addition, while metazoans development relies on complex gene regulatory networks, many of the underlying network motifs (Alon, 2007) are similar or identical to those studied here. Minimal network motifs embedded in larger networks are necessary and sufficient for many network functions, but adding extra connections often adds control, precision and robustness, and may impose its own evolutionary constraints. That said, our work shows that evolutionary constraints already emerge from simple network motifs.

Recent developmental and quantitative genetics studies demonstrated that epistatic interactions among genes are of paramount importance in determining phenotypes (Haag, 2007; Lehner, 2011; Mackay, 2014; Lagator *et al*, 2017a). For example, experiments in *D. melanogaster* have shown that the phenotypic effects of variation in single genes depend heavily on the genetic background, and they do so for phenotypes as varied as bristle shape and number, haltere and eye size, and wing morphology (reviewed in Lehner, 2011; Mackay, 2014). Nonlinear regulatory mechanisms of gene regulation such as those we study are one possible cause of epistatic interactions (Lehner, 2011). To our knowledge, our study shows for the first time that non-additive interactions in a nonlinear gene regulatory network help produce constrained phenotypic variation. These interactions enabled the origin of novel phenotypes that were not observed when we mutated a single gene (e.g. the "flat" phenotype in concurring gradients network, Appendix Fig S6). Our results suggest that these epistatic interactions can also be predicted if the corresponding GRN, its regulatory mechanism and the effect of mutations in single regulatory regions are known. This observation complements a recent study suggesting that epistatic interactions between mutations in transcription factors and DNA binding sites are determined by regulatory network structure (Lagator *et al*, 2017b). Ultimately, understanding the nonlinearities inherent in complex biological systems will be essential to understand how such systems constrain the production of phenotypes.

Since the 19[th] century, Darwinian evolutionary biology has focused on natural selection and its power to shape populations and species. Natural selection, however, requires phenotypic variation, and the molecular mechanisms by which DNA mutations produce novel phenotypes have only become understood in recent years. While orthodox evolutionary theory assumed, often tacitly, that DNA mutations may produce any kind of variation (Mayr, 1963), the discovery of constrained phenotypic variation challenged this view (Smith *et al*, 1985; Arnold, 1992). As we show here, constrained variation in simple yet important spatial gene expression patterns can be explained by the simple fact that genes are embedded in regulatory networks. What is more, the regulatory mechanisms of these GRNs can help explain why specific gene expression patterns originate preferentially. Given the pervasive nonlinearity of gene regulatory networks (Davidson, 2006), we surmise that constraints like those we observe are inherent in biological pattern-forming systems. Future work will show whether they can also influence the trajectories of adaptive evolution.

## Materials and Methods

### Media

Cloning experiments used Luria–Bertani medium (LB: 10 g Bacto-tryptone, 5 g yeast extract, 10 g NaCl per 1 l) supplemented with appropriate antibiotic (100 μg/ml ampicillin, 30 μg/ml kanamycin or 50 μg/ml spectinomycin). Experiments with the complete synthetic circuits used "Stripe Medium" [SM: LB plus 0.4% (w/v) glucose, 50 μg/ml ampicillin, 15 μg/ml kanamycin and 25 μg/ml spectinomycin]. For the opposing gradients network, SM was supplemented with 5 μM isopropyl β-D-1-thiogalactopyranoside (IPTG).

### Molecular cloning reagents

Restriction enzymes and T4 DNA ligase were purchased from New England BioLabs (NEB). Oligonucleotides were obtained from Microsynth and chemicals were obtained from Sigma-Aldrich. Polymerase chain reactions (PCRs) were carried out with KOD Hot Start polymerase (MERCK MILLIPORE). Plasmids were purified using the QIAprep Spin Miniprep Kit (QIAGEN).

### Generation of libraries

Two stripe-forming synthetic circuits of our previous work (Schaerli *et al*, 2014) which implemented the opposing gradients mechanism (GenBank accession codes of plasmids: KM229377, KM229382, KM229387) and the concurring gradients mechanism (GenBank accession codes of plasmids: KM229378, KM229383, KM229388) were used as starting points ("wild-types", WT) of our experiments.

#### Mutating one regulatory region of a circuit at a time

Oligonucleotides covering the regulatory regions of these circuits were synthesised with weighted base mixtures (Isalan, 2006). That is, four custom-weighted phosphoramidite mixtures were prepared, with the WT base pair constituting 95% and each of the other bases constituting 1.67% of any one mixture. These mixtures were used to randomise the regulatory regions (Table 3) during oligonucleotide synthesis (Microsynth). These semi-randomised weighted single-stranded oligonucleotides (2 μM) were annealed to a reverse

**Table 3. Sequences of oligonucleotides synthesised with weighted base mixtures.**

| Network[+] gene | Weighted oligonucleotide for randomisation (5′→3′) |
|---|---|
| OG red | CGTCACACTTTGCTATGCCATAGCATTTTTATCCATAAGATTAGCGGTTCCTACCTGACGCTTTTTATCGCAACTCTCTACTGTTTCTCCATACCGAATTCATTTCACC |
| OG blue | ATTGGAATTCTTTATGGCTAGCTCAGTCCTAGGTACAATGCTAGCGAAGGGTCCCTATCAGTGATAGAGAGAGCTCGTTGAGTTACCTGC |
| OG green | ATTGGAATTCTTGACGGCTAGCTCAGTCCTAGGTACAGTGCTAGCGAAGGGAATTGTTATCCGGATAACAATTCCGAGCTCGTTGAGTTACCTGC |
| CG red | CGTCACACTTTGCTATGCCATAGCATTTTTATCCATAAGATTAGCGGTTCCTACCTGACGCTTTTTATCGCAACTCTCTACTGTTTCTCCATACCGAATTCATTTCACC |
| CG blue | ATTGGAATTCATTTTGGTGACACTATAGAAGGGGCCAAGCAGGGGGCCAAGCAGGGGGCCAAGGAGCTCGTTGAGTTACCTGC |
| CG green | ATTGGAATTCTAATACGACTCACTGTAGGGGAATTGTTATCCGGATAACAATTCCGAGCTCGTTGAGTTACCTGC |
| CG green mutant A | ATTGGAATTCTAATACGACTCACTGTAGGGGAATTGTTAACCGGATAACAATTCCGAGCTCGTTGAGTTACCTGC |
| CG green mutant B | ATTGGAATTCTTAATACGACTCACTGTAGGTGAATTGTTATCCGGATAACATTCCGAGCTCGTTGAGTTACCTGC |

The underlined region was randomised, and the remaining sequence was constant.
OG, opposing gradient network; CG, concurring gradient network.

### Table 4.  Sequences of primers used for cloning.

| Name | Sequence (5′→3′) |
| --- | --- |
| rev_green/blue | GCAGGTAACTCAACGAGCTC |
| rev_red | GGTGAAATGAATTCGGTATGGA |
| pBAD_Gibson_f | TCCATACCGAATTCATTTCACC |
| pBAD_Gibson_r | AGTGTGACGCCGTGCAAATAATC |

primer (Table 4, 2.4 μM) to render them double-stranded by primer extension (2 min 95°C, cooling down to 72°C in 7 min, 10 min 72°C). The resulting library of double-stranded oligonucleotides was then purified with the QIAquick nucleotide removal kit (QIAGEN). Next, these double-stranded oligonucleotides encoding the mutated regulatory regions were cloned into the plasmid encoding the gene whose regulatory region was to be mutated. For the "blue" and "green" regulatory regions, this was done by restriction enzyme digest and ligation. For the "red" regulatory region, Gibson assembly (Gibson *et al*, 2009) was performed instead.

The cloning of the "blue" and "green" regulatory regions by restriction enzyme digest and ligation was performed as follows: double-stranded oligonucleotides encoding one of the regions were digested with EcoRI and SacI. The plasmid into which the region was to be inserted was also digested with EcoRI and SacI, dephosphorylated with CIP and gel purified (QIAquick gel extraction kit, QIAGEN). The double-stranded digested oligonucleotide library (25 ng) was then ligated into the cut plasmid (70 ng).

The cloning of the "red" regulatory regions by Gibson assembly was performed as follows: the plasmid into which the region was to be inserted was first amplified using the primers pBAD_Gibson_f and pBAD_Gibson_r (Table 4), and was then assembled with the double-stranded oligonucleotide library using the Gibson assembly master mix (NEB). Due to a mistake in primer design, Gibson assembly produced a two-nucleotide deletion (CT) downstream of the pBAD promoter. Therefore, the library contains sequences with and without this deletion. We confirmed that this change did not affect the initial "stripe" phenotype.

Ligation ("blue" and "green" regulatory regions) and Gibson assembly reaction products ("red" regulatory region) were transformed into electrocompetent MK01 cells (Kogenaru & Tans, 2014) that carried already the two other plasmids necessary to complete the synthetic circuit. Transformants were plated out on SM-agar plates.

#### Mutating multiple regulatory regions of a circuit

For experiments where multiple regulatory regions of a circuit were to be mutated at the same time, cloning was performed as described above for mutating one regulatory region at a time. However, instead of transforming plasmids with mutagenised regulatory regions directly into MK01 cells, ligation products and Gibson assembly reaction products were first transformed into electrocompetent NEBα cells, and plated out on LB-agar plates containing the appropriate antibiotic (100 μg/ml ampicillin, 30 μg/ml kanamycin or 50 μg/ml spectinomycin). All colonies were resuspended in LB, diluted 100-fold into LB containing the appropriate antibiotic, and grown overnight. Plasmids libraries

were extracted from the resulting culture (QIAprep Spin Miniprep Kit, QIAGEN). The extracted plasmid libraries were mixed with plasmids containing no mutations in a ratio of 70:30 to generate mutant circuits that have mutations in two or three genes. The resulting plasmid mix was transformed into electrocompetent MK01 cells (Kogenaru & Tans, 2014). Circuits that had no mutations or only mutations in one gene were not considered in the analysis.

#### Fluorescence measurements of mutagenised circuits

The agar plates in Fig 1 are only shown to illustrate the spatial pattern formation. The quantitative measurements were all performed in liquid cultures in 384-well plates:

Colonies were picked from agar plates, inoculated into SM medium in a single well of a 96-well plate and grown overnight. Each plate also contained three clones of the WT circuit and a "blank" (SM medium only). A glycerol stock plate was prepared from the overnight cultures. This plate was used to inoculate three further 96-well overnight pre-culture plates with SM medium. Five microliter of each well from the 96-well plate was transferred to four wells of a 384-well plate containing 55 μl of SM medium, arabinose, and IPTG. Specifically, the four wells contained the following amounts of arabinose and IPTG:

1  0% arabinose ("low")
2  0.0002% arabinose ("medium")
3  0.2% arabinose ("high")
4  0.2% arabinose with 700 μM IPTG for the opposing gradients network and 0.2% arabinose with 100 μM IPTG for the concurring gradients network ("metabolic load control", see section "Phenotype classification, metabolic load" for details).

The pipetting steps for this part of the experiment were carried out with a manual pipetting system (Rainin Liquidator 96, METTLER TOLEDO).

The 384-well plate was incubated at 37°C in a Tecan plate reader (Infinite F200 Pro or SPARK 10 M) until the *E. coli* cells had reached stationary phase (~5 h). During this incubation, absorbance at 600 nm and green fluorescence (excitation: 485 nm, emission: 520 nm) were measured every 10 min. Between readings, plates were shaken continually (orbital, 2 mm). Plates were incubated and read with their lids in place to reduce evaporation. Absorbance and green fluorescence were measured for each colony in three independent experiments, each started from a separate pre-culture plate.

#### Analysis

The time-point when the fluorescence of the WT network at the medium arabinose concentration (0.0002%) peaked was chosen for further analysis of all fluorescence measurements. The background fluorescence of the SM medium was subtracted from each culture's fluorescence. Likewise, the background absorbance was subtracted from each culture's absorbance. Background-corrected fluorescence was then normalised for the number of cells by dividing it by the background-corrected absorbance. This background-corrected normalised fluorescence (nF) was used for all further analyses.

All expression data are listed in the file Dataset EV2.

## Exclusions

A circuit was excluded from further analysis if:
1 any of its nF values (except the one at 0% arabinose) was smaller than zero or
2 the absorbance of the circuit differed by more than 0.1 from the absorbance of the WT controls in any of the four conditions (which indicates a substantially different growth rate of the circuit) or
3 it suffered from metabolic load (see below).

### Metabolic load

The 4$^{th}$ condition (highest arabinose concentration with IPTG) served as a metabolic load control. We previously noted that expressing the genes of synthetic networks at high levels can induce a strong bacterial stress response that affects the expression of genes and the growth rate of the cell (Schaerli *et al*, 2014). This can lead to a spurious "stripe" phenotype caused by a stress-induced GFP expression that is shut down at the highest arabinose concentration (Schaerli *et al*, 2014). We therefore checked for each circuit whether it suffers from this metabolic load problem. To this end, we removed LacI repression at the highest of our three arabinose conditions through addition of IPTG. Without repression, we are no longer expecting to observe a "stripe" phenotype. If the observed phenotype is nevertheless a "stripe", this is a strong indication that the network suffers from metabolic load. Specifically, a circuit was excluded due to high metabolic load if its nF value in the 4$^{th}$ condition was lower than 90% of its nF value at the medium arabinose concentration, and if the nF value at the medium arabinose concentration was higher than the corresponding nF of the WT controls [we know that expression levels as high as that of the WT do not induce metabolic load (Schaerli *et al*, 2014)].

## Phenotypic categories

All mutant circuits that remained after the filtering procedure just described were classified into the following phenotypic categories (in this order):

### Broken

A threshold value of nF below which a phenotype was considered "broken" was defined as follows: For the opposing gradients networks, nF needed to lie below the nF of the WT controls at the highest of our three arabinose concentrations. For the concurring gradients networks, nF needed to lie below 1/3 of the nF of the WT controls at this highest arabinose concentration. Any circuit whose nF was below this threshold in all four conditions was assigned the "broken" phenotype. The reason for the different definitions of the threshold for the two networks is that the WT concurring gradient network circuit has a much higher level of basal fluorescence.

### Flat

The average nF of the lowest, medium and highest arabinose concentration was calculated. If all three nFs differed by less than 10% from this average, the circuit was assigned the "flat" phenotype.

### Decrease

If the following three statements were true, the mutant circuit was assigned the "decrease" phenotype:

1 The nF at the lowest arabinose concentration was higher than 90% of the nF at the medium arabinose concentration.
2 The nF at the medium arabinose concentration was higher than 90% of the nF at the highest arabinose concentration.
3 The nF at the lowest arabinose concentration was higher than 120% of the nF at the highest arabinose concentration.

### Increase

If the following three statements were true, the mutant was assigned the "increase" phenotype:
1 The nF at the highest arabinose concentration was higher than 90% of the nF at the medium arabinose concentration.
2 The nF at the medium arabinose concentration was higher than 90% of the nF at the lowest arabinose concentration.
3 The nF at the highest arabinose concentration was higher than 120% of the nF at the lowest arabinose concentration.

### Stripe

If the nF at medium arabinose concentration was higher than 120% of the nF at the lowest and highest arabinose concentrations, the mutant was assigned the "stripe" phenotype.

### Other

Any phenotype that did not fall into one of the previous categories.

## Sequencing

The mutagenised region(s) of all circuits whose phenotype fell into one of our six main categories, and did so consistently in three independent measurements were sent for Sanger sequencing (High-throughput service, Microsynth, see Table 5 for primers).

Mutagenised circuits with WT regulatory sequences despite mutagenesis, with polymorphic nucleotides in the sequences (mainly due to transformation of multiple plasmid variants into the same cell) or with cloning artefacts (shortened or multiple regulatory regions), were discarded. The remaining mutants were used for the phenotypic statistics we report, and their sequences were analysed further. Pairwise alignment to the WT sequence was performed with the Biopython (Cock *et al*, 2009) Bio.pairwise2 module. A custom-made Python script was used to categorise mutations as point mutations, insertions, or deletions, and to identify their positions.

All sequences are listed in Dataset EV1.

## Experimental confirmation of rare observed phenotypes

Phenotypes observed fewer than three times in a library (except "others") were experimentally confirmed. The fluorescence output

**Table 5. Sequences of primers used for sequencing.**

| Name | Sequence (5′→3′) | Used to sequence |
|---|---|---|
| pET_Seq | CCGAAAAGTGCCACCTGAC | OG green |
| pET_Seq_Amp | GACACGGAAATGTTGAATACTCATAC | CG green, OG green |
| pBAD_f | GCCGTCACTGCGTCTTTTAC | OG red, CG red |
| pCDF_Seq_ori | GAGTTCGCAGAGGATTTGTTTAGC | OG blue, CG blue |
| pCDF_I2_rev | TCTACTGAACCGCTCTAG | OG blue |

of these circuits was measured in a 96-well plate assay as described previously (Schaerli *et al*, 2014) at the arabinose concentrations also used in the 384-well plate assay. If the phenotypes of the two assays did not agree, the circuits were excluded from the dataset.

## Experimental confirmation of epistasis

The plasmids from mutant 4_4_f (see Dataset EV1) were isolated (Appendix Fig S6). For Appendix Fig S6A, the mutated plasmid coding for the "green" gene was transformed together with the WT plasmids for the "blue" and "red" genes into electrocompetent MK01 cells (Kogenaru & Tans, 2014). For Appendix Fig S6B, the mutated plasmid coding for the "blue" gene was transformed together with the WT plasmids for the "green" and "red" genes into electrocompetent MK01 cells (Kogenaru & Tans, 2014). For Appendix Fig S6C, the initial 4_4_f mutant was assayed (the "red" gene is not mutated). The fluorescent phenotypes were measured in a 96-well plate assay as described (Schaerli *et al*, 2014) at the following arabinose concentrations (w/v): 0.2, 0.02, 0.002, 0.0002, 0.00002, 0%.

## Statistical tests

Chi-square goodness-of-fit tests (Snedecor & Cochran, 1989) were used to compare observed and expected frequencies. An online tool (https://graphpad.com/quickcalcs/chisquared1.cfm) was used to perform the calculations.

## Phenotype diagrams

A previously developed and experimentally validated model was used to describe the regulatory dynamics of our networks (see Appendix Tables S1 and S2 for details) (Schaerli *et al*, 2014). To generate phenotype diagrams, a custom-made Python script (Code EV2) was written that systematically varies the combinations of two or more parameters between 0–110% (for the "blue" and "green" genes) or 0–200% (for the "red" genes) of the wild-type parameter value in 1,000 steps. Analogously to our experiments for each parameter combination, the model's phenotype was evaluated at three arabinose concentrations (0, 0.000195, 0.2% for opposing gradients network and 0, 0.000195, 0.1% for concurring gradients network). In order to allocate the obtained GFP expression pattern to a phenotype category, the same rules as described above for the experimental data ("Phenotypic categories") were used. An R script (R Development Core Team, 2016; Code EV2) was applied to create a digital image where every pixel corresponds to a combination of parameter values and has a colour corresponding to the model's phenotype. For Appendix Fig S3, combinations of two parameters as indicated on the axes were varied. For Fig 3A, multiple parameters affecting the promoter or operator were varied jointly and to the same extent. For the opposing gradients network, these were parameters $a$ and $b$ for promoter activity, and parameter $c$ for operator activity. For the concurring gradients network, these were parameters $b$, $c$ and $e$ for promoter activity and parameter $d$ for operator activity.

## Distributions of novel phenotypes

In order to fit quantitatively the distributions of novel phenotypes for the single-gene mutants and predict the distributions of novel phenotypes for the multiple-gene mutants, a custom-made Python script (Code EV3) was used. We first discuss the single-gene mutants:

Multiple iterations of a procedure were performed that consisted of the following three steps: first, a series of simulated mutants was created. For each gene, a random binary vector whose length corresponds to the number of nucleotides in the regulatory sequence (Appendix Table S4) was generated. In this binary vector, 0's and 1's indicate whether a nucleotide is not mutated (0) or mutated (1), and the probability to obtain either 0 or 1 is given by the average mutation rate extracted from the experimental sequencing data (Appendix Table S4). For every network, it was then assessed which genes were mutated, according to the entries of this vector. For the single-gene mutants, mutants that had only one gene mutated were selected. This process was repeated until 1,000 single-gene mutants had been obtained. For a particular mutant, all parameters related to the mutated sequence were varied (see Appendix Tables S1–S3). A given single-gene mutant had either only its promoter mutated, only its operator mutated, or both. If a mutation affected the promoter (operator), all parameters determining promoter (operator) activity were changed (Appendix Table S3). A subset of the changed parameters was varied jointly and to the same extent (i.e. all of them were changed to same percentage of their wild-type parameter value), because a mutation is likely to affect these parameters in a similar way (Appendix Table S3). New (mutant) parameters were chosen according to a standard uniform distribution between upper and lower ranges which were kept constant for a given model iteration.

Second, for each mutant the phenotype of the model (Appendix Tables S1 and S2) was evaluated at three arabinose concentrations (0, 0.000195, 0.2 for opposing gradients network and 0, 0.000195, 0.1 for concurring gradients network). In order to allocate the resulting GFP expression pattern to a phenotype category, the same rules as described above for the experimental data ("Phenotypic categories") were used.

Third, the obtained phenotype distribution of the 1,000 assessed mutants was compared to the experimentally observed phenotype distribution.

After each iteration of these three steps, the upper and lower ranges of each parameter were manually adjusted to best fit the results of the model to the phenotype distributions observed in the experimental data (see Appendix Discussion of Lower and Upper Bounds of the Parameter Intervals). Finally, the best upper and lower ranges for each mutant parameter distribution were kept (Appendix Table S3) and used to produce Fig 2C.

To predict the distributions of novel phenotypes for the multiple-gene mutants (Fig 2E), the same procedure was used, with the following modifications: (i) only mutants containing more than one mutated gene were kept for analysis. (ii) The upper and lower ranges of the parameter distributions were not adjusted, but the intervals derived from the single-gene mutants were used (Appendix Table S3). (iii) Only one iteration of the three steps above was performed.

## Schematic drawings

Figure 3B contains schematic depictions, based on the mathematical steady-state model. The model uses the parameters as in Appendix Tables S1 and S2 with following changes:

Figure 3B: opposing gradient (left): Parameter $c$ of the "green" gene was changed from 0.103 to 0; concurring gradient (right): Parameter $c$ of the "green" gene was changed from 16.5 to 0.

## Data availability

The plasmids of the starting networks (Schaerli *et al*, 2014) are available on the GenBank (https://www.ncbi.nlm.nih.gov/genbank/) with following access codes: KM229377, KM229382, KM229387 (opposing gradients network) and KM229378, KM229383, KM229388 (concurring gradients network). Sequences of the regulatory regions of all the mutants reported in this study are in Dataset EV1. Measured expression levels of all the mutants reported in this study are in are in Dataset EV2. Scripts used to generate the expression dynamics, the phenotype diagrams and the distribution of phenotypes are provided as Codes EV1, EV2 and EV3, respectively.

**Expanded View** for this article is available online.

## Acknowledgements

We thank Elke Karaus Wyer for carrying out preliminary experiments and Joshua L. Payne for critical reading. We thank the Sanger Sequencing team from Microsynth for their support. YS, JMD and LM acknowledge support by the Swiss National Science Foundation (PZ00P3-148235 and 31003A_175608 to YS); AW acknowledges support by Swiss National Science Foundation grant 31003A_146137, by ERC Advanced Grant 739874, by an EpiphysX RTD grant from SystemsX.ch and by the University Priority Research Program in Evolutionary Biology at the University of Zurich. MI is funded by a Wellcome Trust UK New Investigator Award (WT102944) and by the Volkswagen Foundation. AJ and JS acknowledge the Spanish Ministry of Economy, Industry and Competitiveness (MINECO), BFU2010-16428, the European Union's Horizon 2020 research and innovation program under Grant Agreement No. 670555, the European Union Seventh Framework Program (FP7/2007-2013) under grant agreement 601062; the Spanish Ministry of Economy, Industry and Competitiveness "Centro de Excelencia Severo Ochoa 2013–2017", SEV-2012-0208 and the Cerca Programme/Generalitat de Catalunya.

## Author contributions

YS, AW designed the project; YS, JMD, LM performed experiments; YS, AJ, AW analysed data; AJ, YS, JR did the mathematical modelling; JS supervised modelling; YS, AJ, MI, JS, AW wrote the paper.

## Conflict of interest

The authors declare that they have no conflict of interest.

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
