## [Review Process File · Molecular Systems Biology]

Synthetic circuits reveal how mechanisms of gene regulatory networks constrain evolution

Yolanda Schaerli, Alba Jiménez, José M. Duarte, Ljiljana Mihajlovic, Julien Renggli, Mark Isalan, James Sharpe, Andreas Wagner.

Review timeline:

Initial Submission date:	13 th November 2017
Editorial Decision:	4 th January 2018
Appeal:	18 th January 2018
Editorial Decision:	31 st January 2018
Resubmission received:	30 th April 2018
Editorial Decision:	20 th June 2018
Revision received:	2 nd July 2018
Editorial Decision:	8 th August 2018
Revision received:	15 th August 2018
Accepted:	15 th August 2018

Editor: Maria Polychronidou

Transaction Report:

1st Editorial Decision

4th January 2018

Thank you again for submitting your work to Molecular Systems Biology. We have now heard back from the three referees who agreed to evaluate your manuscript. As you will see below, the reviewers raise substantial concerns on your work, which unfortunately preclude its publication at Molecular Systems Biology.

The reviewers acknowledge that the addressed topic is potentially interesting. However, they point out that as it stands the study remains rather preliminary and does not provide sufficiently conclusive quantitative and mechanistic insights. As such, they rated the conclusiveness and the conceptual advance as "Medium/Low" and indicated that they do not support publication at Molecular Systems Biology.

REFERE REPORTS.

Reviewer #1:

Schaerli et al address a very timely and relevant problem of biology in general, namely how molecular mechanisms interact with evolutionary forces to shape phenotypes. They employ synthetic biology ideas to build two different synthetic transcriptional networks that display a stripe-like phenotype when the bacterial lawn is grown under changing levels of an inducer (arabinose). While the two networks have very different topologies, they display a similar phenotype - the stripe. Upon mutagenesis of different promoter regions of the two networks they build libraries of various

networks which show very different stripe-like phenotypes, which are interpreted as an example of constraints on evolution, and many parallels are drawn to developmental systems in terms of GRN topologies and phenotypes.

The way the data is presented and analyzed is to me rather preliminary. There is a qualitative result of constrained evolution, and the only 'mechanism' one can infer from the way the data is presented is that the two networks differ in their topology. This to me is not enough to qualify as a molecular mechanism. The model is not presented in any detail, and thus hard to understand what it incorporates and what it does. There is a surprising lack of a null model, namely a model to which one should compare one's expectations on what the two networks should do. We are simply told that the two networks differ in their distribution of observed phenotypes upon mutation and this is the sole evidence for the conclusion that is spelled out in the title itself.

The frequent and all pervasive parallels and comparisons to GRN's in development of insects is speculative at best in my opinion. While Monod at his time was right to say that "what is true for *E. coli* is true for an elephant", I think especially given what we have learned about synthetic circuits and how fickle they are to engineer, I would be very careful in making too many and too strong comparisons with development, especially when one compares a very early insect embryo that is one big cell with thousands of nuclei to a lawn of *E. coli* with millions of individual bacteria where there is zero communication between cells, to take just one example. Furthermore, as the authors themselves point out, only one of the synthetic networks they built has a topological analog in development, whereas the second one "to the best of our knowledge the concurring gradients mechanism has so far not been observed in a natural stripe-forming regulatory network".

When comparing the experimental data to the model data there is a lack of any type of quantitative analysis and the only thing we are left with is to use is to rely on the human eye: lines 385-386. In fact the old human eye shows that there are some significant differences between fig 2d and fig 2e.

Thus while I think that the question is a great one, the combo of experiment and model needs to be restructured and tightened a lot more in order for a coherent and solid story to emerge. The work could benefit from the old adage of "Less is more", i.e. go in more depth in a few cases of the mutated networks which would allow one to be actually quantitative, as opposed to presenting a statistical aspect of phenotypes, where one has little control over the molecular details. In addition, the writing can greatly benefit from being concise and sticking to the facts and not trying to make too far flung connections to developmental circuitry, as after all we are dealing with lawns of bacteria and the scales and many molecular mechanisms are vastly different, despite the fact that a basic very much stripped down (in case of the developing embryo) GRN topology is similar to the synthetic one.

Reviewer #2:

In the manuscript "The mechanism of gene regulatory networks constrain evolution: A lesson from synthetic circuits," Schaerli and colleagues explore evolutionary biases using two synthetic gene regulatory circuits expressed in *E. coli* that produce a gene expression stripe. They show that mutations cause distinct phenotypes in the two networks using a combination of experiments and modeling, suggesting that network structure can constrain the outcome of evolution. These results are perhaps not surprising from a theoretical perspective, but as the authors point out experimental validation is needed.

Taken together I found this to be an interesting paper addressing an important question. The paper is well written and a pleasure to read. However, I have some recommendations.

1) At the end of the first results paragraph, the authors mention how the mechanism is not observed in natural systems. Could they comment in the discussion on why? Has their synthetic system taught us anything novel about this? They start to discuss this, but it would be nice to be more explicit and in depth.

2) It appears the authors binned the different point mutations. It would be interesting to see if the SNPs versus INDELS varied in the magnitude of their effects (or not).

3) The authors binned phenotypes into six categories: "increase," "decrease," etc. While this makes the data accessible to visualization, I would like to see how different mutations vary in their magnitude and how network topology may influence this. Treating this data with a more quantitative approach could have important consequences for the interpretation of the data.

Taken together, this is a nice manuscript. But a deeper look into the data, specifically in the magnitude of changes, and how this is influenced by network topology could be informative.

Reviewer #3:

The manuscript presents a study of mutations of synthetic gene networks, designed to produce a "stripe" phenotype. The idea is to compare two different architectures producing the same phenotype, to mutate them, and to compare to models. The first interesting aspect of the paper are the data produced by this very original experimental approach in evo-devo (and I would recommend to make all of them accessible with the article, in a format easy to use and to analyze).

It is then found that the two network structures give different phenotypes when mutated "in the same way" (statistically speaking). This by itself is not surprising to me (see below); however what is very surprising is that the statistics of the mutated phenotypes found experimentally are essentially very consistent with simple models of mutated gene networks, which in my opinion is one of the most interesting results of the paper.

As said above, I thus found the paper rather interesting and innovative. However in my opinion there are issues with the way results are put forward and described, I feel some explanations are missing, especially on the theory side.

Major comments:

* The aspect I found most frustrating in the main text is the lack of clear explanation on why the distribution of mutated phenotypes can be explained by the model. In Supplement, it is suggested that network topology matters more than parameters for the phenotypes, which provides a reasonable explanation for this effect. I think this should really be clarified and put forward in the main text if true. Indeed, looking at details, many mutations seem consistent with broad "loss of function" effects; for instance after a loss of function of the "red" gene, I would expect a priori an "increase" phenotype since it would effectively move the positional information of the network leftwards, and this is what is observed. This is very important to give more explanations on such effects from a theoretical standpoint, since this confirms that even simple models can be predictive of evolution and mutation.

* Conversely, I do not think the qualitative conclusion put forward that the same phenotype implemented in the different way would lead to different novel phenotypes once mutated is really surprising in any way. I understand this study is empirical but it is performed on a synthetic system designed to perform precisely as predicted by the simple theory. But if I have an activation via double repression vs via double activation (which is what happens for the green gene), of course I expect very different mutated phenotypes. Such ideas have even been studied extensively by Michael Savageau in his "demand theory". In short, Savageau has compared networks with double repression vs activation and argued that some types of networks are favoured because of their difference in mutated phenotypes. Even though this was not done in an evo-devo context, I believe some citation and discussion of those ideas are important (for a review of Savageau's work the authors can check Chapter 11 of Uri Alon's Systems Biology book). Savageau even proposed theoretical approaches to project phenotype space on a low dimension spaces (see e.g. Savageau et al, "Phenotypes and tolerances in the design space of biochemical systems", PNAS, 2009) which appear quite similar to some methods used here (e.g. on Fig 3).

* I am not sure the paper is readable as is for colour blind people. This is not a minor issue since the authors label genes and constantly refer to them by their colour in the main text. I would strongly advise to double check this, and, if needed, to label the genes in a different way (using names or

letters).

Minor comments:

* when I first read Figure 2, I did not really understand what "model" meant. Fig 2 c and e are cited only very late in the text, it is a bit strange since there is hardly any comment on Fig 2 when the model is introduced. I feel that Fig 2c and e could be put separately late in the text when the mutations are discussed.

* It took me some time to understand what authors precisely meant by "novel" phenotypes, it is essentially "not stripe", it should be clarified

* Fig 1c, right panel: I found a bit surprising that the "green" gene is drawn "above" the blue gene. Since the blue gene is activating the green gene, this suggests some kind of "amplification" mechanisms, is it correct? Is it observed experimentally as well or is it a misleading figure?

* On the experimental side, if I understand correctly, the original manuscript really implements a spatial gradient (as illustrated in Fig 1), while the current study is looking at uniform input but at different levels. This raises the question of the effect of spatial diffusion in both the original and current study, is there any reason to think that it does not matter? This is important for the current study because diffusion might actually "kill" some "stripe" phenotypes if a real spatial gradient were to be implemented.

* I feel some more reference to earlier work would be helpful on top of Savageau. For instance, I think it would be fair to refer to the original works detailing existing stripe modules, rather than to more recent reviews. As far as I know, the first "stripe" module very similar to the "opposing gradient" model was suggested for *eve-2* in 1991 by Stanojevic et al in *Science*. Regarding evolution and constrained variation on non-linear networks that is mentioned in discussion, there are also interesting earlier work, e.g. Francois et al, *Molecular Systems Biology* 2007 which precisely performs evolutionary simulations of stripes and studies how it biases future evolution (e.g. they even get the opposing gradient system).

Appeal

18th January 2018

Many thanks for sending the reviewer's comments for our study "The mechanisms of gene regulatory networks constrain evolution: A lesson from synthetic circuits" (MSB-17-8102).

We were pleased with the overall enthusiasm for the questions we are addressing and our approach to it. Comments include "very timely and relevant problem of biology" (reviewer 1), "interesting paper addressing an important question" (reviewer 2), "very original experimental approach in evo-devo" (reviewer 3).

We are thus very surprised that our paper was rejected, given that all reviewers gave some very positive comments about the paper. They simply would like to see some more analyses and discussion of the data. We found the specific suggestions constructive and we will be able to address them. Notably, none of the reviewers asked for more experiments. Here is a summary of the main points the reviewer would like to see:

Reviewer 1:

- Null model
- Compare experimental and model in a quantitative way

Reviewer 2:

- SNPs vs INDELS
- Analysis of the phenotypes in a more quantitative way

Reviewer 3:

- Explain why the model fits the data well
- Better discussion of previous work
- Readable for colour-blind people

Adding these suggested analyses and discussing the mentioned literature will substantially strengthen our study and allow us to emphasize the conclusiveness and the conceptual advance of our work. Based on the very positive feedback that we received from many colleagues on our preprint, we are confident that this will be a highly cited publication.

Therefore, we formally request an appeal. If given the opportunity for revision, we can easily perform the extra analyses, make these points of conceptual advance much clearer, as well as addressing all the minor points raised by the referees. We are thus keen to write a point-by-point rebuttal sincerely addressing all of the points in a measured manner.

We do hope that you will grant our appeal and allow us to submit this revision.

2nd Editorial Decision

31st January 2018

Thank you for your message regarding our decision on your manuscript MSB-17-8102. I have now read once again your manuscript and the referee reports and I have carefully considered the points raised in your appeal letter.

During the review of your work, the referees were not convinced that the study offers sufficient quantitative and mechanistic insights and they rated the conceptual advance and the conclusiveness of the work as "Medium/Low". In particular, reviewer #1 thought that the study remains quite preliminary and mentioned that it provides limited mechanistic insights and lacks quantitative comparisons between experimental data and the model predictions. During our 'pre-decision cross-commenting' process (according to which we circulate the reports to all reviewers and give them the chance to make comments), Reviewer #3 mentioned that s/he agrees with the comments of reviewer #1's indicating that quantitative analyses are required to potentially reveal actual mechanistic insights. Overall, the lack of quantitative analyses/insights was a prominent concern raised and as reviewer #2 points out "a more quantitative approach could have important consequences for the interpretation of the data". Considering that addressing these concerns would require substantial additional analyses with unclear outcome and in combination with our editorial policy is in principle to allow a single round of major revision, we do not see strong reasons for reverting our initial decision.

However, we recognize that the reviewers did have positive words for the questions addressed in the study and for the approach used. As such, we would be willing to consider a new and extended manuscript based on this work, provided that the issues raised by the reviewers, including those emphasized above, can be convincingly addressed. As I mentioned, we recognize that this would involve substantial further analyses, and as you probably understand we can give no guarantee about its eventual acceptability. If you do decide to follow this course then it would be helpful to enclose with your re-submission an account of how the work has been altered in response to the points raised by the reviewers.

At this point, I am afraid I see no choice but to confirm our negative decision. I hope that the comments above can better explain the reasons behind our decision and that they will be helpful for you in deciding how to proceed with the study.

Resubmission - authors' response

30th April 2018

Reviewer #1:

Schaerli et al address a very timely and relevant problem of biology in general, namely how molecular mechanisms interact with evolutionary forces to shape phenotypes.

We thank the reviewer for this positive comment.

They employ synthetic biology ideas to build two different synthetic transcriptional networks that display a stripe-like phenotype when the bacterial lawn is grown under changing levels of an inducer

(arabinose). While the two networks have very different topologies, they display a similar phenotype - the stripe. Upon mutagenesis of different promoter regions of the two networks they build libraries of various networks which show very different stripe-like phenotypes, which are interpreted as an example of constraints on evolution, and many parallels are drawn to developmental systems in terms of GRN topologies and phenotypes.

The way the data is presented and analyzed is to me rather preliminary. There is a qualitative result of constrained evolution, and the only 'mechanism' one can infer from the way the data is presented is that the two networks differ in their topology. This to me is not enough to qualify as a molecular mechanism.

The proof that the two different networks use a distinct dynamical regulatory mechanism is provided in detail in our previous publication (Nat. Commun., 5:4905). In particular, Figure 1c of that previous publication shows the phase portraits of the different networks, demonstrating that they are using qualitatively different regulatory mechanisms to yield the stripe phenotype.

We now extended Fig.1c in the current publication to show the different spatiotemporal courses of gene expression for the two mechanisms and also extended following sentence in the text (p4): “We previously built multiple 3-gene synthetic networks that display the same gene expression phenotype, but create this phenotype through different regulatory mechanisms (Schaerli et al., 2014) – different regulatory dynamics and regulatory interactions among network genes resulting in different spatiotemporal gene expression profiles (Cotterell and Sharpe, 2010; Jimenez et al., 2015; Schaerli et al., 2014).” We would also like to point out that in Box1 we explain the regulatory mechanisms of the two networks in detail.

The model is not presented in any detail, and thus hard to understand what it incorporates and what it does.

The model was described in detail in the methods and the supporting information. However, we agree, that there was not enough description of the model in the main text. We now added Table 1 containing the equations for the “green” genes and explaining the biological meaning of each parameter. We also explain the model better in the main text. E.g. we moved the discussion of the upper and lower bounds for the distribution from which the “mutated” parameters are drawn from the supporting information into the main text under the heading “Phenotype distributions can be explained by the model”.

There is a surprising lack of a null model, namely a model to which one should compare ones expectations on what the two networks should do. We are simply told that the two networks differ in their distribution of observed phenotypes upon mutation and this is the sole evidence for the conclusion that is spelled out in the title itself.

The point is well taken. One can distinguish between two different kinds of null models here. The first is a phenomenological null model that does not incorporate any assumptions about the architecture of a GRN. The most parsimonious such null model is one where each non-stripe phenotype is produced at the same frequency, and these frequencies are identical for the two networks investigated. Our data are clearly inconsistent with it (Fig. 2). The second kind of null model would assume some sort of “default” architecture of a stripe-forming GRN and derives a distribution of novel phenotypes from it. However, such a default architecture does not exist, and even if it did exist, it is not clear that a synthetic circuit with such an architecture could be built. We therefore chose to compare two circuits that can be built against each other. So, if you will, one circuit serves as a null model to create an expected phenotype distribution for comparison to the phenotype distribution produced by the other circuit. We agree that we did not make this rationale clearer and now do so as follows in the discussion:

“Both networks produced a non-uniform distribution of novel phenotypes and are thus inconsistent with a naïve expectation (null model) that each non-stripe phenotype is produced at the same

frequency. More interestingly, the different networks displayed different phenotypic variation and consequently different constraints in the production of novel phenotypes.“

We respectfully disagree with the reviewer that we do not have any explanation for the constrained variation of phenotypes that we observed. The evidence that the observed constraints can be explained by the underlying regulatory mechanisms is presented in Figs. 3- 6. We show that the simple model capturing the regulatory mechanisms of the circuits is enough to reproduce the experimentally observed phenotypic variation. Moreover, the model predictions are supported by our sequencing data. In other words, we have experimental data, a mathematical model that not only explains this data, but also provides a mechanistic explanation, and further sequencing data that support the model. Taken together, we think that this is sufficient evidence for our conclusion that the regulatory mechanisms of GRNs constrain the phenotypic variation produced.

The frequent and all pervasive parallels and comparisons to GRN's in development of insects is speculative at best in my opinion. While Monod at his time was right to say that "what is true for *E. coli* is true for an elephant", I think especially given what we have learned about synthetic circuits and how fickle they are to engineer, I would be very careful in making too many and too strong comparisons with development, especially when one compares a very early insect embryo that is one big cell with thousands of nuclei to a lawn of *E. coli* with millions of individual bacteria where there is zero communication between cells, to take just one example. Furthermore, as the authors themselves point out, only one of the synthetic networks they built has a topological analog in development, whereas the second one "to the best of our knowledge the concurring gradients mechanism has so far not been observed in a natural stripe-forming regulatory network".

We fully agree that a lawn of *E. coli* does not capture the complexity of a developing animal. But it is exactly this reduced complexity that allowed us to study the potential of GRNs to bias phenotype production without confounding effects. We mention this in the discussion: “And while a lawn of *E. coli* cells carrying a synthetic GRN does not capture the complexity of a developing animal, this reduced complexity also allowed us to study the potential of GRNs to bias phenotype production without confounding effects.”

Following your advice, however, we now removed the following sentence from our manuscript and the corresponding discussion from the supporting information: “We emphasize that phenotypes like these are not peculiarities of our synthetic system. They have also been observed in developing organisms, for example in the anterior–posterior patterning of mutant fly embryos (*Drosophila melanogaster* and *Megaselia abdita*) (see SI discussion “Biological examples of our phenotype categories in fly embryos”)”.

It is true, that the to the best of our knowledge the concurring gradients mechanism has so far not been observed in a natural stripe-forming regulatory network. However, previous studies added this network to the repertoire of possible stripe-forming mechanisms (Munteanu et al., 2014; Rodrigo and Elena, 2011; Schaefer et al., 2014).(We added this now in the text on p.6). More importantly, our main conclusion that the regulatory mechanisms of networks restrict the possible phenotypic variation upon mutation is independent of whether the studied mechanisms have already been observed in a natural stripe-forming regulatory network or not.

When comparing the experimental data to the model data there is a lack of any type of quantitative analysis and the only think we are left with is to use is to rely on the human eye: lines 385-386. In fact the old human eye shows that there are some significant differences between fig 2d and fig 2e.

We have now converted the initial Figs. 2b/c into Table 2. Initial Figs 2d/e are now in Fig. 6 and the underlying data in SI Table 5. This allows for a quantitative analysis. We performed Chi-square tests comparing the experimental to the model data. They indicate no significant difference ($p > 0.05$) between the experimental and model phenotype distributions for the “blue” and “green” genes when mutating a single gene. Likewise, Chi-Square tests indicate no significant difference between the experimental and model data when mutating multiple genes (opposing: [Chi-square, $X^2(5, N = 36)$

= 6.962, $p = 0.2235$]; concurring: [Chi-square, $\chi^2(5, N = 41) = 5.552, p = 0.3522$]). We include these tests now in the text. We also added a paragraph where we explain in detail why the biggest difference between the experimental data and the model is in the phenotype category “others”. “The most significant difference between the phenotype distributions of the experiments and the model is the fraction of networks displaying phenotypes in the “other” category, which is consistently higher in the model predictions than in our experimental data. This can be explained by the fact that we excluded any network from further analysis that displayed phenotypes falling into different categories in at least one of three replicate phenotype measurements (see Methods for details)...”

Thus while I think that the question is a great one, the combo of experiment and model needs to be restructured and tightened a lot more in order for a coherent and solid story to emerge. The work could benefit from the old adage of "Less is more", i.e. go in more depth in a few cases of the mutated networks which would allow one to be actually quantitative, as opposed to presenting a statistical aspect of phenotypes, where one has little control over the molecular details.

Following the reviewer’s suggestion, we go into greater depths in our analysis of the green gene (Figs. 3+4) and present pertinent quantitative data.

In addition, the writing can greatly benefit from being concise and sticking to the facts and not trying to make too far flung connections to developmental circuitry, as after all we are dealing with lawns of bacteria and the scales and many molecular mechanisms are vastly different,

despite the fact that a basic very much stripped down (in case of the developing embryo) GRN topology is similar to the synthetic one.

Following the reviewer’s suggestion, we have tried to do this.

Reviewer #2:

In the manuscript "The mechanism of gene regulatory networks constrain evolution: A lesson from synthetic circuits," Schaeferli and colleagues explore evolutionary biases using two synthetic gene regulatory circuits expressed in *E. coli* that produce a gene expression stripe. They show that mutations cause distinct phenotypes in the two networks using a combination of experiments and modeling, suggesting that network structure can constrain the outcome of evolution. These results are perhaps not surprising from a theoretical perspective, but as the authors point out experimental validation is needed.

Taken together I found this to be an interesting paper addressing an important question. The paper is well written and a pleasure to read. However, I have some recommendations.

We thank the reviewer these positive comments and the recommendations which we address below:

1) At the end of the first results paragraph, the authors mention how the mechanism is not observed in natural systems. Could they comment in the discussion on why? Has their synthetic system taught us anything novel about this? They start to discuss this, but it would be nice to be more explicit and in depth.

In our previous publication (Nat. Commun., 5:4905), we showed experimentally that all four incoherent feedforward loops (I1-I4) can form a stripe. To the best of our knowledge, only the I1 and I2 (opposing gradients) networks have so far been reported to be part of natural stripe-forming networks. It is indeed an intriguing question, why the I3 (concurring gradients) and I4 networks are less abundant or even not found in developing systems. This question has previously been discussed in the literature, for example in

Rodrigo, G., and Elena, S.F. (2011). Structural discrimination of robustness in transcriptional feedforward loops for pattern formation. PLoS One 6, e16904.
and

Munteanu, A., Cotterell, J., Sole, R.V., and Sharpe, J. (2014). Design principles of stripe-forming motifs: the role of positive feedback. *Sci Rep* 4, 5003.

In the current study, we did not focus on the question of why a specific network design is chosen during evolution, but we showed that alternative designs (once evolved for whatever reason) will be distinct in how they constrain the outcome of further evolution.

We state this now in the discussion: “Why a specific GRNs and not one its alternatives evolves remains an open and important question (Carroll, 2008)..... our observations show that seemingly equivalent solutions to solve a biological problem are also distinct on what qualitatively novel phenotypes (which might be adaptive) are accessible.”

2) It appears the authors binned the different point mutations. It would be interesting to see if the SNPs versus INDELS varied in the magnitude of their effects (or not).

Our library was designed to contain SNPs only. Due to the imperfect oligo synthesis we also have a low fraction (<5%) of INDELS. However, their numbers are too low to make any statistically sound conclusion of whether their effects are different from SNPs, especially as mutants usually carry SNPs and INDELS at the same time.

We clarify now our library composition in following sentence on p.9: “Resulting average mutation rates per regulatory regions ranged from 2.6 to 3.5 mutations (mainly point mutations and <5% of insertions and deletions) per regulatory region with individual mutants carrying 1 to 9 mutations (SI sequences, SI Table 4).”

3) The authors binned phenotypes into six categories: "increase," "decrease," etc. While this makes the data accessible to visualization, I would like to see how different mutations vary in their magnitude and how network topology may influence this. Treating this data with a more quantitative approach could have important consequences for the interpretation of the data.

We thank the reviewer for this suggestion. We now prepared a new figure (Fig. 2c) that examines the magnitude of the expression level changes by comparing the GFP expression level of the individual mutants at medium arabinose concentration to the GFP expression levels at low (x axis) and high arabinose (y axis) concentrations. We note that the previously classified phenotypes (Fig. 2a) form well-separated clusters in this analysis. For example, networks in the bottom-right quadrant correspond to “stripe” phenotypes, because their pattern is described as an increase (positive x-axis) followed by a decrease (negative y-axis) in expression. Consequently, “decrease” and “increase” phenotypes occupy the upper-right and bottom-left quadrants, respectively. We also sequenced the mutated regulatory regions of all analysed networks, and find a weak association between the number of mutations a network carries, and the extent to which its observed phenotype differs from the starting stripe phenotype (as quantified through the Euclidean distance) (SI Figure 2). However, it is possible to get non-stripe phenotypes even with single point mutations (shown as red arrows in Fig. 4b, and SI Fig. 5).

Taken together, this is a nice manuscript. But a deeper look into the data, specifically in the magnitude of changes, and how this is influenced by network topology could be informative.

We agree with this general suggestion, and hope the reviewer agrees that we have now put significant extra effort into achieving a deeper look – which has indeed been beneficial to the paper overall.

Reviewer #3:

The manuscript presents a study of mutations of synthetic gene networks, designed to produce a "stripe" phenotype. The idea is to compare two different architectures producing the same phenotype, to mutate them, and to compare to models. The first interesting aspect of the paper are the data produced by this very original experimental approach in evo-devo (and I would recommend to make all of them accessible with the article, in a format easy to use and to analyze).

We thank the reviewer for these positive comments. We now provide all the sequences (SI sequences) and the expression data (SI expression) to the reader as Supporting information.

It is then found that the two network structures give different phenotypes when mutated "in the same way" (statistically speaking). This by itself is not surprising to me (see below); however what is very surprising is that the statistics of the mutated phenotypes found experimentally are essentially very consistent with simple models of mutated gene networks, which in my opinion is one of the most interesting results of the paper. As said above, I thus found the paper rather interesting and innovative. However in my opinion there are issues with the way results are put forward and described, I feel some explanations are missing, especially on the theory side.

Major comments:

* The aspect I found most frustrating in the main text is the lack of clear explanation on why the distribution of mutated phenotypes can be explained by the model.

We have now extended the description of the model and moved parts from the supporting information into the main text (under the subheading "Phenotype distributions can be explained by the model"). We explain there that we simulate mutations by replacing the wildtype parameters with an activity drawn from a uniform distribution and aimed to identify upper and lower bounds for this distribution to give the best possible agreement between the experimental and model data. We also discuss the thus identified bounds and their biological meaning (please see also our reply to your next but one comment).

In Supplement, it is suggested that network topology matters more than parameters for the phenotypes, which provides a reasonable explanation for this effect. I think this should really be clarified and put forward in the main text if true.

We now moved this figure into the main text (Fig 5) and added a whole section describing it ("Regulatory mechanisms influence the phenotype distributions more than the actual parameters of the network").

Indeed, looking at details, many mutations seem consistent with broad "loss of function" effects; for instance after a loss a function of the "red" gene, I would expect a priori an "increase" phenotype since it would effectively move the positional information of the network leftwards, and this is what is observed. This is very important to give more explanations on such effects from a theoretical standpoint, since this confirms that even simple models can be predictive of evolution and mutation.

Yes, indeed, most observed phenotypes are consistent with decreasing a parameter value ("loss of function"). This is reflected by the fact that most of the identified upper bounds of the "mutated" parameters lie below the wild-type, unmutated values. We discuss this now in detail on p.21: "For 20 out of 25 parameters the lower and upper bounds of the identified intervals were equal or below the WT parameter value (=100%), consistent with the notion that most random mutations will disrupt molecular interactions and thus decrease the corresponding parameter's value, which is also in agreement with available mutagenesis data of the components used in our synthetic circuits (Imburgio et al., 2000; Niland et al., 1996; Shin et al., 2000). ..."

* Conversely, I do not think the qualitative conclusion put forward that the same phenotype implemented in the different way would lead to different novel phenotypes once mutated is really surprising in any way. I understand this study is empirical but it is performed on a synthetic system designed to perform precisely as predicted by the simple theory. But if I have an activation via double repression vs via double activation (which is what happens for the green gene), of course I expect very different mutated phenotypes. Such ideas have even been studied extensively by Michael Savageau in his "demand theory". In short, Savageau has compared networks with double repression vs activation and argued that some types of networks are favoured because of their difference in mutated phenotypes. Even though this was not done in an evo-devo context, I believe some citation and discussion of those ideas are important (for a review of Savageau's work the authors can check Chapter 11 of Uri Alon's Systems Biology book). Savageau even proposed theoretical approaches to project phenotype space on a low dimension spaces (see e.g. Savageau et al, "Phenotypes and tolerances in the design space of biochemical systems", PNAS, 2009) which appear quite similar to some methods used here (e.g. on Fig 3).

We thank the reviewer for pointing us to this relevant literature. Indeed, Savageau proposed that transcriptional regulation by an activator or by a repressor can lead to different phenotypes upon mutation. Depending on whether a regulated gene usually needs to be highly or lowly expressed, the error-load caused by a mutation is different when the gene is regulated positively or negatively, favouring one or the other mode of regulation.

We think our work complements Savageau's work because his original idea focuses on maintaining the initial regulation, whereas our work focuses on what qualitatively novel phenotypes (which might be adaptive) are accessible by a network. Another difference is that we are working with non-linear systems and looking at expression level patterns in a concentration gradient, which are more difficult to capture in a simple demand rule.

However, we do agree that it is important to discuss Savageau's work in our paper. We now cite his work in the introduction and discuss it in more detail in the discussion in the following context: "Why a specific GRNs and not one its alternatives evolves remains an open and important question (Carroll, 2008). In an attempt to understand the rules that govern this selection, Savageau formulated its "demand rule" (Savageau, 1977). He observed that activators and repressors can achieve the same regulatory goals, but that frequently expressed genes tend to be regulated by activators (positive mode), whereas rarely expressed genes tend to be regulated by repressors (negative mode). These differences can be explained by the fact that the negative and positive regulatory modes can lead to different phenotypes upon mutation and the error-load caused by a mutation is different, favouring one or the other mode of regulation (Savageau, 1977, 1983, 1998a, b). While Savageau's work focuses on maintaining the initial regulation, our observations show that seemingly equivalent solutions to solve a biological problem are also distinct on what qualitatively novel phenotypes (which might be adaptive) are accessible."

* I am not sure the paper is readable as is for colour blind people. This is not a minor issue since the authors label genes and constantly refer to them by their colour in the main text. I would strongly advice to double check this, and, if needed, to label the genes in a different way (using names or letters).

We do take accessibility for colour-blind people seriously. Indeed, one of our authors is colour-blind and the paper is readable for him. Nevertheless, we now chose slightly different shades of red, blue and green that are unambiguous to all colour blind people (according to <http://jfly.iam.u-tokyo.ac.jp/color/index.html#pallet>)

Minor comments:

* when I first read Figure 2, I did not really understand what "model" meant. Fig 2 c and e are cited only very late in the text, it is a bit strange since there is hardly any comment on Fig 2 when the model is introduced. I feel that Fig 2c and e could be put separately late in the text when the mutations are discussed.

We agree that this was a weak point of our manuscript. We chose this layout to prevent that the reader has to compare data across figures. We now chose to present the data of the original Fig. 2b/c as table and Fig. 2d/e as Figure 6. We also redisplay some data to prevent that the reader has to compare data across figures.

* It took me some time to understand what authors precisely meant by "novel" phenotypes, it is essentially "not stripe", it should be clarified.

We now clarify on p.5: "Here we go beyond this question to ask whether different GRNs that have the same phenotype (a "stripe" of gene expression) can produce different novel (*i.e.*, "non-stripe") gene expression phenotypes..."

* Fig 1c, right panel: I found a bit surprising that the "green" gene is drawn "above" the blue gene. Since the blue gene is activating the green gene, this suggests some kind of "amplification" mechanisms, is it correct? Is it observed experimentally as well or is it a misleading figure?

Yes, it is correct that there is some expected amplification. Each T7 RNA polymerase (blue gene) can produce multiple mRNAs coding for GFP. Please also refer to the experimental data in our initial publication (Nat. Commun., 5:4905).

* On the experimental side, if I understand correctly, the original manuscript really implements a spatial gradient (as illustrated in Fig 1), while the current study is looking at uniform input but at different levels. This raises the question of the effect of spatial diffusion in both the original and current study, is there any reason to think that it does not matter? This is important for the current study because diffusion might actually "kill" some "stripe" phenotypes if a real spatial gradient were to be implemented.

In our initial publication we tested both – a spatial gradient on agar plates as well as different input concentrations in well-plates. We did not observe any difference between the two assay systems. All the quantitative data presented in the paper was based on measurements in well- plates. Our network components are all intracellular and do not diffuse between cells. As far as we could observe, the arabinose concentration gradient on the agar plates is stable during the time course of our experiments.

* I feel some more reference to earlier work would be helpful on top of Savageau. For instance, I think it would be fair to refer to the original works detailing existing stripe modules, rather than to more recent reviews. As far as I know, the first "stripe" module very similar to the

"opposing gradient" model was suggested for *eve-2* in 1991 by Stanojevic et al in Science. Regarding evolution and constrained variation on non-linear networks that is mentioned in discussion, there are also interesting earlier work, e.g. Francois et al, Molecular Systems Biology 2007 which precisely performs evolutionary simulations of stripes and studies how it biases future evolution (e.g. they even get the opposing gradient system).

Thanks for suggesting these references. We are now citing them in the introduction.

Thank you for submitting your manuscript, related to your previous submission MSB-18-8102. We have now heard back from the reviewers #2 and #3 who agreed to evaluate your manuscript. As you will see below, the reviewers mention that most of their concerns have been satisfactorily addressed.

However, reviewer #2 refers to the need to include some further discussion and clarifications. We would ask you to address these issues in a minor revision.

Before we formally accept the manuscript for publication, we would also ask you to address a few remaining editorial issues listed below.

REFEREE REPORTS.

Reviewer #2:

In their revised manuscript Schaerli and colleagues have done an excellent job sharpening up their manuscript. For example, I found the addition of the tables highlighting the formula has brought clarity to the manuscript. The addition of the quantitative data to the manuscript - i.e., figure two - is appreciated.

I am still disappointed by the lack of quantitative data or how it is being presented. In figure 1 it appears that the GFP signal from the opposing gradient versus concurring gradient looks patchy in expression. Is this real or is it due to the nature of the assay? What is the timescale for figure 1C? For those that come from different biological systems, this would be a great place to note this. It seems possible to explore these properties and how they shape the evolvability - something that the authors have thought deeply about and that would benefit the system/paper. The authors should at least discuss some of these points, which have been explored in depth by the synthetic biology community.

Finally, I think the manuscript would benefit from an additional paragraph in the discussion explicitly addressing the differences in *E. coli* versus metazoans. For example, what are the timescale differences in *E. coli* versus multicellular animals (related to my comment above towards)? This does not have to be a detraction from the author's system, instead, it can be a way to discuss potential differences, caveats, and what we can learn from these reduced systems (again this could be a strength).

Reviewer #3:

The authors have taken into account my comments and I am happy with the current revision.

2nd Revision - authors' response

2nd July 2018

Point-by-point response to the remaining reviewer concerns

Reviewer #2:

In their revised manuscript Schaerli and colleagues have done an excellent job sharpening up their manuscript. For example, I found the addition of the tables highlighting the formula has brought clarity to the manuscript. The addition of the quantitative data to the manuscript - i.e., figure two - is appreciated.

Thank you for recognising that the quality of our manuscript improved.

I am still disappointed by the lack of quantitative data or how it is being presented. In figure 1 it appears that the GFP signal from the opposing gradient versus concurring gradient looks patchy in expression. Is this real or is it due to the nature of the assay?

Yes, the patchiness is due to the nature of this agar plate-based assay. Any small irregularity in the growth of the cell lawn will be detected also in the fluorescence channel. Please note, however, that we show these plates only to illustrate the spatial pattern formation. The quantitative measurements were all performed in liquid cultures in 384-well plates where we do not have this problem, as we now mention on page 31 line 671.

What is the timescale for figure 1C? For those that come from different biological systems, this would be a great place to note this.

The time needed for stripe formation in the agar plate assay is 6 hours (after addition of arabinose). We now provide this information in the figure legend. The schematics in Fig. 1C represents modelling data, so the timescale there refers to arbitrary units.

In our last publication (Schaerli et al.; Nat. Commun. 2014; Fig. 4), we also measured experimentally the temporal dynamics of stripe formation in growing *E. coli* cells and showed a qualitative agreement with the predictions. We also mention this fact now in the manuscript on p.7, line 148 (new text highlighted in red):

“Fig. 1c schematically shows the temporal expression profiles of the three genes and their steady-state profiles (last panel) of the three genes (color-coded as in Fig. 1a) under varying arabinose concentrations, as previously determined experimentally (Schaerli et al., 2014).”

It seems possible to explore these properties and how they shape the evolvability - something that the authors have thought deeply about and that would benefit the system/paper. The authors should at least discuss some of these points, which have been explored in depth by the synthetic biology community.

We did not study whether the speed of stripe formation affects the evolvability of the gene regulatory network. This is indeed an interesting question and is a possible direction for future research.

Finally, I think the manuscript would benefit from an additional paragraph in the discussion explicitly addressing the differences in *E. coli* versus metazoans. For example, what are the timescale differences in *E. coli* versus multicellular animals (related to my comment above towards)? This does not have to be a detraction from the author's system, instead, it can be a way to discuss potential differences, caveats, and what we can learn from these reduced systems (again this could be a strength).

We added the following text highlighted in red to the discussion on p. 25, starting on line 548:

“And while a lawn of *E. coli* cells carrying a synthetic GRN does not capture the complexity of a developing animal, this reduced complexity also allowed us to study the potential of GRNs to bias phenotype production without confounding effects. In addition, while metazoans development relies on complex gene regulatory networks, many of the underlying network motifs (Alon, 2007) are similar or identical to those studied here. Minimal network motifs embedded in larger networks are necessary and sufficient for many network functions, but adding extra connections often adds control, precision and robustness, and may impose its own evolutionary constraints. That said, our work shows that evolutionary constraints already emerge from simple network motifs.

Reviewer #3:

The authors have taken into account my comments and I am happy with the current revision.

We thank the reviewer for his or her support, and for taking the time to review our paper again.

Comment to editorial issues:

- Due to the quantitative nature of the study we would encourage you to provide the Source Data for the Figures showing essential quantitative information (e.g. Figure 2). Source Data should be provided in a single .zip folder labeled "Source Data for Figure 2" and including a README.txt file briefly explaining the content of the folder/subfolders.

All the data used to produce all the experimental figures is contained in the Datasets EV1 and EV2. Moreover, the data of Figure 2 is also listed in Table 2 and the data of Fig. 6b/c is listed in Appendix Table S5.

4th Editorial Decision

8th August 2018

Thank you for sending us your revised manuscript. Before we formally accept it for publication, we would ask you to address a few remaining issues listed below.

We are offering a "model curation service" (still in a pilot phase) together with Prof. Jacky Snoep and the FAIRDOM team. In brief, the aim is to enhance reproducibility and add value to papers including mathematical models. Prof. Snoep's summary on the model (*Model Curation Report*) is pasted below. As he has already discussed with you, there are some issues, which we would ask you to fix when you submit your revision. I hope you agree that the model curation is a useful initiative and we would of course love to hear your feedback or suggestions.

****Model Curation Report**:**

In their manuscript: "The mechanisms of gene regulatory networks constrain evolution: A lesson from synthetic circuits", the authors present simulation results for two regulatory networks, resulting in either an "opposing" or "concurring gradient". Most of the simulation results are for steady state solutions and the steady state equations, with complete parameter sets, are given in the manuscript. Although it is possible to find the original ODE models for the systems in a cited paper by Mangan and Alon, and in a previous paper by the authors, I do recommend to give a full description of the ODEs and the steady state solutions in the supplementary information of the current manuscript. In figure 1c dynamic results are shown and to reproduce these the reader must have access to the ODE models. The models are also very small so it is easy enough to fully describe them in the S.I.

When I tried to reproduce the simulations in Figure 1, I obtained similar results, but could not precisely reproduce the figures. Upon contacting the authors I learnt that the figures are "schematic depictions", i.e. they are based on model simulations but are not direct simulation results. The authors chose to show the schematic results as they better convey the message that the authors want to get across. In itself I think this is OK, but I would recommend the authors to show the original simulations in Supplementary Information.

The authors have sent the models as Matlab files, which I will use to code the dynamic models and make SBML versions available.

4th Revision - authors' response

15th August 2018

We thank Prof. Snoep for his useful feedback. We made now following changes:

- We include now the full description of the ODEs in the Appendix (Appendix Model, equations 1-9). The steady state solutions were already present in Appendix Tables S1 and S2.
- We mention in the caption of Fig. 1c that these are schematic drawings and refer to Appendix Figures S7 for the simulations.
- We include the ODE simulations in Appendix Figure S7.
- We provide the corresponding matlab script as Computer Code EV1 (please note, that we introduced few minor changes concerning the degradation rate parameters compared to the version we sent to Prof. Snoep).

MOLECULAR SYSTEMS BIOLOGY

Corresponding Author Name: Yolanda Schaeferli & Andreas Wagner
 Manuscript Number: MSB-17-8102RR